# Training Data Size Induced Double Descent For Denoising Feedforward Neural Networks and the Role of Training Noise

**Rishi Sonthalia**                                           *rsonthal@math.ucla.edu*
*Department of Mathematics*
*University of California, Los Angeles*

**Raj Rao Nadakuditi**                                      *rajnrao@umich.edu*
*Department of EECS*
*University of Michigan, Ann Arbor*

**Reviewed on OpenReview:** *https://openreview.net/forum?id=FdMWtpVT1I*

## Abstract

When training an unregularized denoising feedforward neural network, we show that the generalization error versus the number of training data points is a double descent curve. We formalize the question of how many training data points should be used by looking at the generalization error for denoising noisy test data. Prior work on computing the generalization error focuses on adding noise to target outputs. However, adding noise to the input is more in line with current pre-training practices. In the linear (in the inputs) regime, we provide an asymptotically exact formula for the generalization error for rank 1 data and an approximation for the generalization error for rank $r$ data. From this, we derive a formula for the amount of noise that needs to be added to the training data to minimize the denoising error. This results in the emergence of a shrinkage phenomenon for improving the performance of denoising DNNs by making the training SNR smaller than the test SNR. Further, we see that the amount of shrinkage (ratio of the train to test SNR) also follows a double descent curve.

## 1 Introduction

Denoising noisy training data is a widely used technique for pretraining networks to learn good data representations. Two extremely common examples of pretraining via denoising are Masked Language Modelling (MLM) (Devlin et al., 2019) and Stacked Denoising Autoencoders (SDAE) (Vincent et al., 2010). For many modern problems, we work at large scales in terms of the number of parameters and the number of training samples. Recently there has been significant work in understanding the effect of scaling the number of parameters in a neural network. This resulted in the discovery of the much celebrated double descent phenomena (Belkin et al., 2019). However, we have a weaker understanding of the effect of scaling the number of data points. Classical works such as Krogh & Hertz (1991); Geman et al. (1992); Opper (2002) and more recent work such as Gerace et al. (2020); Nakkiran et al. (2020); Nakkiran (2020); d'Ascoli et al. (2020); Adlam & Pennington (2020) show either empirically or via theoretical analysis that sample wise double descent exists. However, these were in the regime of supervised learning. On the other hand, our motivation comes from understanding denoising neural networks. For MLM and SDAEs, denoising is a pretraining procedure, in which case the generalization error would depend on the downstream task. As a first step, we shall instead look at the generalization error for denoising test data. The difference between the prior supervised learning setup and our denoising setup can be seen in Figure 1.

To understand the denoising setting, we empirically show that sample-wise double descent exists for denoising feedforward neural networks (Section 3). We show that shrinking the training data Signal to Noise Ratio (SNR) (i.e., increasing the amount of training data noise) for fixed test data SNR can mitigate this double

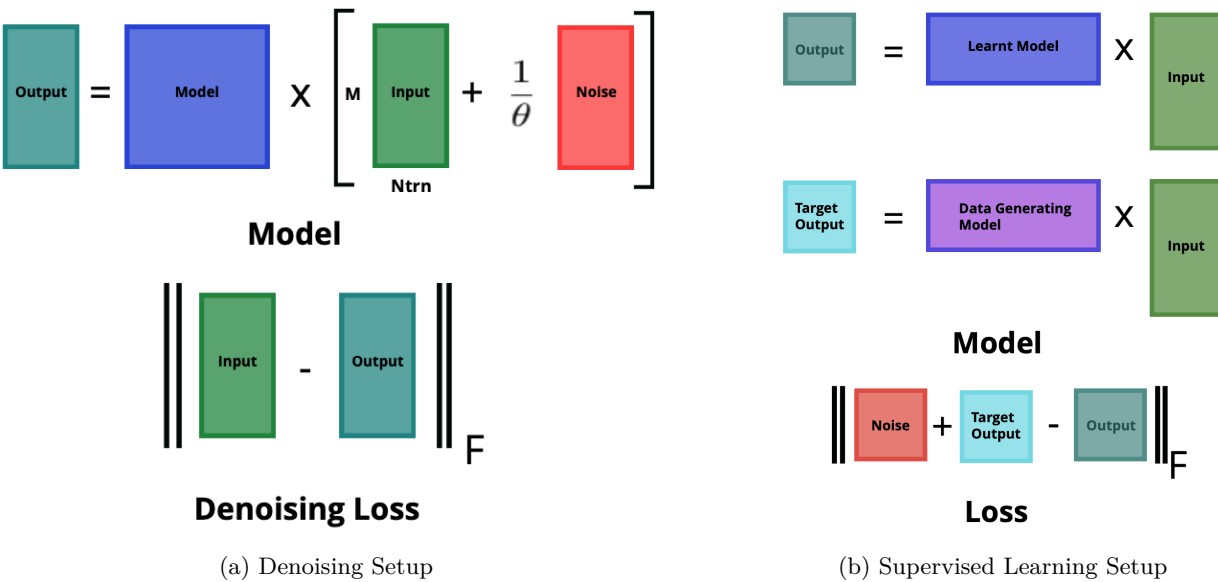

(a) Denoising Setup        (b) Supervised Learning Setup

Figure 1: Figure showing the difference in the noise placement between the traditional supervised learning setup for which empirical and theoretical double descent curves have been found versus our denoising setup for which we recover double descent curves.

descent. Moreover, we show that the curve for the ratio of the best training data SNR to the test data SNR also has sample-wise double descent (Section 3). To theoretically understand the phenomena, we look at the simplest setting. Specifically, we look at the case when we have a one-layer linear network, and we are denoising data that lies on a line embedded in high dimensional space (Section 4). In this setting, we derive the exact asymptotics for the generalization error (Section 5). We show that the generalization error and optimal training noise level spike at the interpolation threshold. From the theoretical analysis, we see that the spike occurs due to the variance of the model increasing. We use the rank one result to derive an approximation for general (low) rank $r$ data.

**Contributions.**

The main contributions of the paper are as follows.

1. We empirically show that when denoising data using a feedforward network, the curve for the generalization error versus the number of training data points $Ntrn$ and the curve for the ratio of the test data SNR to the optimal training data SNR has double descent. Further changing the training data SNR can mitigate the double descent in the generalization error curve. Thus the noise level acts as a regularizer.
2. Assuming we have mean zero, bounded variance, and rotational invariant noise, we derive an analytical formula for the expected mean-squared generalization error for denoising data that lives in a one-dimensional linear subspace by a linear model. Further, we use the same method to present a heuristic for higher-rank data and experimentally determine the formula's accuracy for general low-rank data.
3. We show that sample-wise double descent exists for the generalization error and the amount of noise that should be added, even in this simple model.

## 1.1 Related work

Understanding deep neural networks is a currently active area of research with many exciting theoretical results. The discovery that fixed depth infinite width (under certain limits) neural networks can be thought of as kernel regression (Jacot et al., 2018) and the discovery of double descent for neural networks (Belkin et al., 2019) has sparked significant research into understanding the generalization error in the linear regime

(in parameters, not inputs). The exact asymptotic for the generalization error was first understood for ridge regression (Bartlett et al., 2020; Hastie et al., 2022; Belkin et al., 2020; Advani & Saxe, 2020; Mel & Ganguli, 2021; Dobriban & Wager, 2018). Following this, many papers have studied the situation for the Random Features model and the Neural Tangent Kernel (NTK) model (Jacot et al., 2020; Mei & Montanari, 2019; Ghorbani et al., 2021; Adlam & Pennington, 2020; Geiger et al., 2020). Other recent work for supervised learning includes work on multiple descents (Derezinski et al., 2020; d'Ascoli et al., 2020; Liang et al., 2020), transfer learning (Lampinen & Ganguli, 2019), and Gaussian mixture models (Loureiro et al., 2021). However, to our knowledge, there has yet to be any work that looks at the problem for the denoising setup.

The idea of adding noise to improve generalization has been seen before. One popular strategy is to use Dropout (Hinton et al., 2012; Wan et al., 2013; Srivastava et al., 2014), where we randomly zero out either neurons or connections. Another idea that is commonly used is data augmentation. In a revolutionary paper, Krizhevsky et al. (2012) showed that augmenting the dataset with noisy versions of the images greatly improved the accuracy. Another area where noise is useful is adversarial learning. Dong et al. (2021) shows epoch-wise double descent for adversarial training. In recent theoretical work related to SDAEs, Pretorius et al. (2018) derived the learning dynamics of a linear autoencoder in the presence of noise. They also establish some relationships between the noise added and weight decay. However, they do not look at the generalization error or quantify the optimal amount of noise that should be added. Gnansambandam & Chan (2020) looked at the problem of determining the optimal amount of noise that should be added. However, they studied this from the perspective of minimizing the variance of the generalization error.

Additionally, there has been significant progress in understanding the Bayes optimal solution when denoising via matrix factorization (Lelarge & Miolane, 2017; Lesieur et al., 2017; Maillard et al., 2022; Troiani et al., 2022; Nadakuditi, 2014). It is important to note that these works do not think of the noise as a regularizer and do not consider the effect of noise on parametric models such as neural networks.

## 2 Problem Set Up

Our goal is to understand the impact of training noise on the generalization error in the context of denoising neural networks. Concretely, suppose we have access to noisy data $y_1, \ldots y_{N_{tst}} \in \mathbb{R}^M$ such that $y_i = \theta_{tst} x_i + \xi_i$, where $x_i \in \mathbb{R}^M$ is sampled from an unknown data distribution $\mathcal{D}$, $\xi_i \in \mathbb{R}^M$ is sampled from some noise distribution $\mathcal{D}_{noise}$, and $\theta_{tst} \in \mathbb{R}$ is a known *scalar* which controls or models how noisy the data is. We study the classic denoising problem of recovering $x_i$ from $y_i$ (James & Stein, 1992; Wiener, 1949; Banham & Katsaggelos, 1997; Benesty et al., 2010; Takeda et al., 2007; Buades et al., 2005). One approach to solving this problem is to learn a function that removes the noise from a set of examples, for instance, using a neural network (Tian et al., 2020). To this end, suppose the noise distribution $\mathcal{D}_{noise}$ is known, then given noiseless data $x_1^{trn}, \ldots, x_{N_{trn}}^{trn}$ we can create noisy versions $y_i^{trn} = \theta_{trn} x_i^{trn} + \xi_i^{trn}$ of our training data. Now consider a neural network denoted $f$, which is trained to minimize the following $\ell_2$ loss function

$$\ell(f; x_i^{trn}) = \frac{1}{N_{trn}} \sum_{i=1}^{N_{trn}} \|x_i^{trn} - f(\theta_{trn} x_i^{trn} + \xi_i^{trn})\|^2. \tag{1}$$

We are then interested in the following mean squared generalization error.

$$\frac{1}{N_{tst}} \sum_{i=1}^{N_{tst}} \|x_i^{tst} - f(\theta_{tst} x_i^{tst} + \xi_i^{tst})\|^2. \tag{2}$$

**The major question we want to answer is the following. Given noisy test data, such that $\theta_{tst}$ is known, what is the optimal value of $\theta_{trn}$ such that a neural network trained using the loss function in Equation 1, minimizes the generalization error in Equation 2? We are also interested in the effect the number of training data points $N_{trn}$ has on the optimal $\theta_{trn}$.**

## 2.1 Signal to Noise Ratio (SNR)

A quantity of interest to us will be the SNR. To properly account for this, if $\mu_{data}$ is the expected norm of the data points and $\mu_{noise}$ is the expected norm of the noise vectors, then we shall call

$$\hat{\theta}_{trn} := \frac{\theta_{trn}\mu_{data}}{\mu_{noise}}, \text{ and } \hat{\theta}_{tst} := \frac{\theta_{tst}\mu_{data}}{\mu_{noise}}$$

to be the training and test data signal to noise ratios.

## 3 Empirical Double Descent

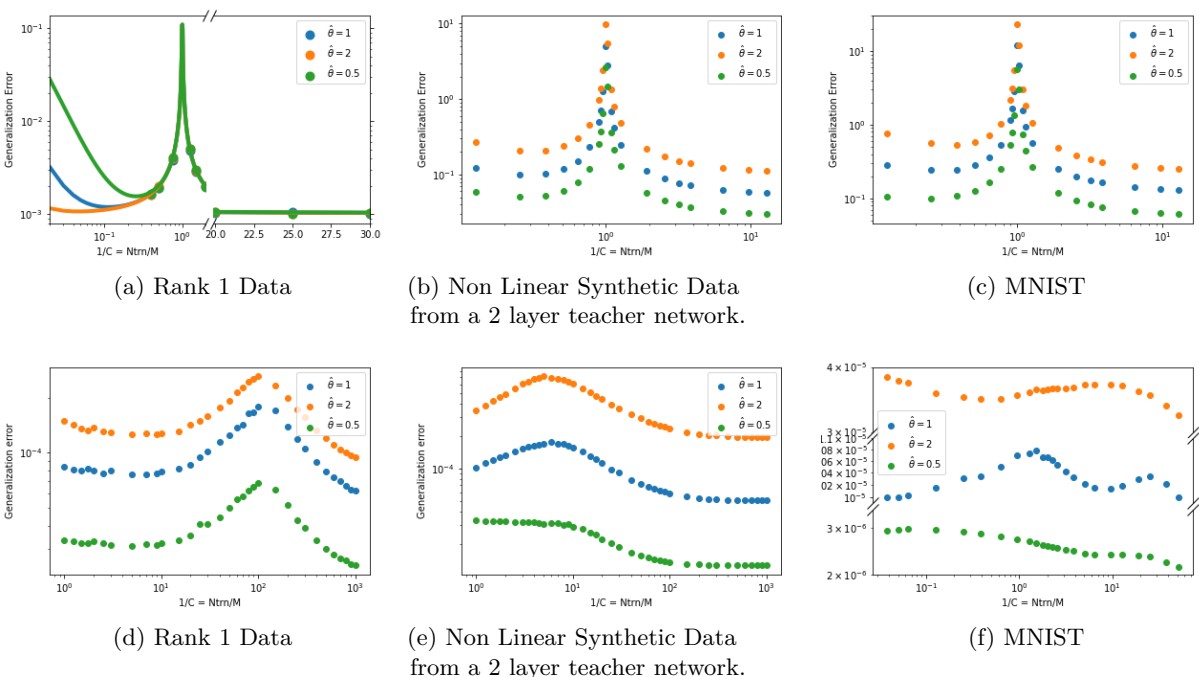

(a) Rank 1 Data

(b) Non Linear Synthetic Data from a 2 layer teacher network.

(c) MNIST

(d) Rank 1 Data

(e) Non Linear Synthetic Data from a 2 layer teacher network.

(f) MNIST

Figure 2: Figure showing the empirical double descent phenomena for the generalization error versus $1/C$ (Number of training samples $Ntrn$ /number of features $M$). The top row is for a linear (with respect to the inputs) network, and the bottom row is for a three-layer ReLU network with width equal to the dimension of the data. The networks were trained with the mean squared denoising error. Here the training data SNR and the test data SNR both equal $\hat{\theta}$. The solid line in Figure (a) is our theoretical line from Theorem 1.

We run two experiments to better empirically understand the interaction between $\theta_{tst}, N_{trn}, \theta_{trn}$ and the generalization error (Eq. 2). First, we show that sample-wise double descent occurs for denoising neural networks empirically. That is, if we fix $\theta_{tst}, \theta_{trn}$, then as we vary $N_{trn}$, we get that the $N_{trn}$ versus generalization error curve has double descent. Second, we explore the role of the amount of training noise and show that optimally picking $\theta_{trn}$ can mitigate the previously seen double descent.

### 3.1 Double Descent for Denoising Networks

For our first experiment, we show that the $N_{tst}$ versus generalization error curve has double descent in simple cases. To do this, we train two feedforward networks (one-layer and three-layer) on three different datasets. The first data set is a line embedded in high-dimensional space. The second data set is a synthetic dataset using a teacher network. That is, the data is generated by sampling latent variables from a Gaussian distribution and then using the outputs from a randomly initialized untrained 2-layer neural network as our data. Finally, the third dataset is MNIST.

Figure 2 shows that if we train a (one-layer and three-layer) feedforward network to denoise data such that the training data signal to noise ratio (SNR) $\hat{\theta}_{trn}$ is the same SNR as that of the test data set ($\hat{\theta}_{tst}$), then double descent occurs in the curve for the denoising generalization error vs. the number of training samples. However, unlike other hyperparameters, such as the number of features and the number of training epochs, we cannot arbitrarily change the number of data points as we are limited by our data set. Hence it could be the case that our maximum number of data points corresponds to the peak of the generalization error curve. To get around this, we can look at the amount of noise we add to the training data. Note that we could have also added other forms of regularization, but the noise level is a natural hyper-parameter here.

## 3.2 Role of Training Noise Level

To see the effect of training data SNR $\hat{\theta}_{trn}$, for a variety of different ratios $\hat{\theta}_{trn}/\hat{\theta}_{tst}$, we compute the denoising generalization error versus the number of data points curve. Figures 3a and 3b show that if we optimally pick the ratio $\hat{\theta}_{trn}/\hat{\theta}_{tst}$, then double descent can be mitigated. We do this for the MNIST and CIFAR datasets. We create test data by taking the test data for each and then adding Gaussian noise. We fix the test SNR $\hat{\theta}_{tst}$ to be 1 for both datasets. Hence we know the test data SNR. We then take various different fractions of the training data and train a three-layer ReLU neural network (without bias) for various levels of training data SNR $\hat{\theta}_{trn}$. For each pair of parameters (number of training data points and the level of training noise), we compute the generalization error averaged over twenty trials for MNIST and five trials for CIFAR. Here the test noise and training noise are resampled for each trial. The plots for the generalization error can be seen in Figures 3a (MNIST) and 3b (CIFAR10), and the plots for the optimal ratio can be seen in Figure 4.

We see five interesting and exciting phenomena from this experiment.[1]

1. For most values of the ratio $\hat{\theta}_{tst}/\hat{\theta}_{trn}$, we see sample-wise double descent for the generalization error.
2. We see that the optimal denoising error does not occur when the train SNR is equal to the test SNR. We need to shrink the train SNR (i.e., increase the test to train SNR ratio). This shrinkage is reminiscent of other shrinkage phenomena such as James & Stein (1992); Tibshirani (1996); Nadakuditi (2014).
3. As seen in Figures 4a and 4b, the optimal ratio depends on the number of data points.
4. Figure 4 shows that the curve for the best $\hat{\theta}_{trn}/\hat{\theta}_{tst}$ also has sample wise double descent.
5. Picking the optimal amount of noise can mitigate sample-wise double descent of the generalization error. This mitigation is reminiscent of how optimal regularization can mitigate double descent in the supervised setting (Nakkiran et al., 2020).

We postulate that spike in generalization error is due to the variance of the model increasing. Hence when we increase the amount of noise, we implicitly regularize the model (Bishop, 1995). This increased regularization results in a decrease in the variance and improves the generalization error.

# 4 Theoretical Problem Assumptions

To be able to provide a theoretical understanding of the five phenomena discovered in Section 3, we consider a simple model that can be theoretically analyzed.

## 4.1 Assumptions about the data

First, we detail assumptions about the data generation process. Specifically, we assume that the data lies in some low-dimensional linear space.

**Assumption 1.** *Let $U \in \mathbb{R}^{M \times r}$ such that the columns of $U$ have unit norm and are pairwise orthogonal. To generate data, we sample latent variables $V^T \in \mathbb{R}^{r \times N}$ and $\Sigma \in \mathbb{R}_+^{r \times r}$ such that $V$ has columns that have unit norm and are pairwise orthogonal and $\Sigma$ is a diagonal matrix with non-negative entries on the diagonal such that $\|\Sigma\|_F = 1$. Then a data matrix $X$, in which each* column *is a data point, is given by $X = U\Sigma V^T$.*

---

[1]Other forms of regularization could remove some of these features. However, we look at the effect of the level of noise by itself.

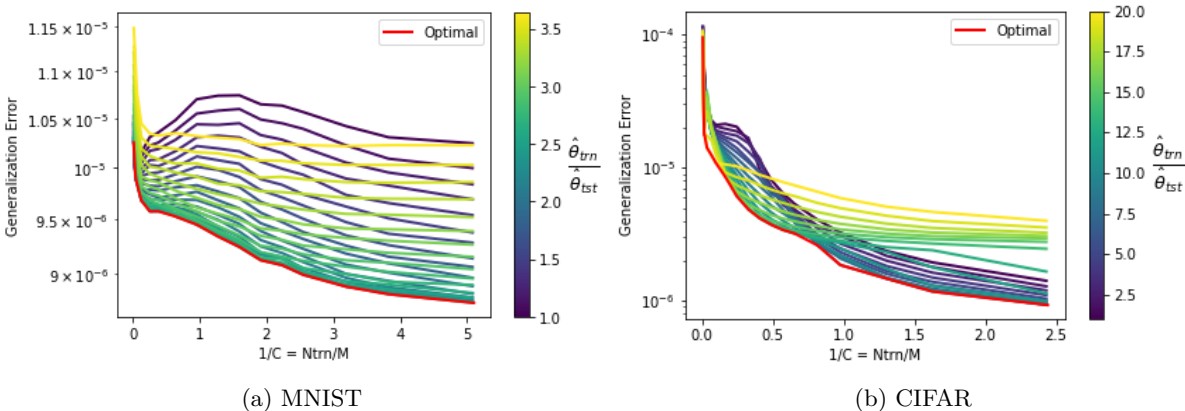

(a) MNIST            (b) CIFAR

Figure 3: Figure showing the empirical denoising generalization error for a three-layer neural network with the width the same as the dimension of the data trained for various different values of $\hat{\theta}_{trn}/\hat{\theta}_{tst}$ and number of training data points. Each neural network was trained for 1500 epochs, using MSE loss and gradient descent with a learning rate of $10^{-3}$. For MNIST, we averaged over twenty trials, and for CIFAR10, we averaged over five trials.

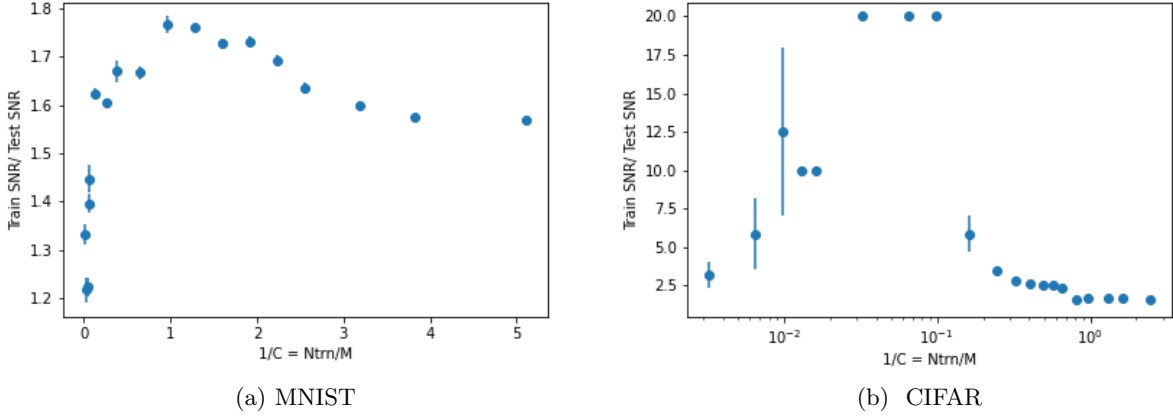

(a) MNIST            (b) CIFAR

Figure 4: Figure showing the sample-wise double descent for the empirically optimal amount of training noise. The figure displays the optimal $\theta_{tst}/\theta_{trn}$ ratio seen empirically versus $1/c = N_{trn}/M$. The ratios plotted here correspond to the ratio for the red line in Figure 3.

Hence, we see that we generate data that lives in a dimension $r$ subspace. Note that we make no assumptions about the distribution of the latent variables $V$ or $\Sigma$. We have two matrices, $X_{trn}$ and $X_{tst}$, corresponding to the train and test data sets. Hence we have corresponding $V_{trn}^T \in \mathbb{R}^{r \times N_{trn}}$, $V_{tst}^T \in \mathbb{R}^{r \times N_{tst}}$, and $\Sigma_{trn}, \Sigma_{tst}$. We make no other assumptions on $U, V_{trn}, \Sigma_{trn} V_{tst}, \Sigma_{tst}$.

### 4.2 Assumptions about the noise

Next, we detail our assumptions about the noise added to the data. For that, we need the following definitions.

**Definition 1.** *A matrix $Z \in \mathbb{R}^{m \times n}$ sampled from a distribution is rotationally bi-invariant if for all orthogonal $U_1 \in \mathbb{R}^{m \times m}$ and all orthogonal $U_2 \in \mathbb{R}^{n \times n}$, $U_1 Z U_2$ has the same distribution as $Z$.*

Another way to phrase rotational bi-invariance is if $A = U_A \Sigma_A V_A^T$ is the SVD, then $U_A$ and $V_A$ are uniformly random orthogonal matrices and are independent of $\Sigma_A$ and each other.

**Definition 2.** *Let $c \in (0, \infty)$ be a shape paramter. Then the Marchenko Pastur distribution with shape $c$ is the measure $\mu_c$ supported on $[c_-, c_+]$, where $c_{\pm} = (1 \pm \sqrt{c})^2$ is such that*

$$\mu_c = \begin{cases} \left(1 - \frac{1}{c}\right) \delta_0 + \nu & c > 1 \\ \nu & c \leq 1 \end{cases}$$

*where $\nu$ has density*

$$d\nu(x) = \frac{1}{2\pi xc} \sqrt{(c_+ - x)(x - c_-)}.$$

With these definitions, we have the following assumptions about the noise matrices $A_{trn}, A_{tst}$.

**Assumption 2.** *Let $A \in \mathbb{R}^{M \times N}$ such that $A$ is sampled from a distribution $\mathcal{D}_{noise}$ such that*

1. *For all $i, j$, $\mathbb{E}_{\mathcal{D}_{noise}}[A_{ij}] = 0$.*

2. *For all $i, j$, $\mathbb{E}_{\mathcal{D}_{noise}}[A_{ij}^2] = 1/M$.*

3. *For all $i_1, i_2, j_1,$ and $j_2$ such that $i_1 \neq i_2$ or $j_1 \neq j_2$, we have that $A_{i_1 j_1}$ and $A_{i_2 j_2}$ are uncorrelated. That is, $\mathbb{E}_{\mathcal{D}_{noise}}[A_{i_1 j_1} A_{i_2 j_2}] = \mathbb{E}_{\mathcal{D}_{noise}}[A_{i_1 j_1}] \mathbb{E}_{\mathcal{D}_{noise}}[A_{i_2 j_2}]$*

4. *$A$ is rotationally bi-invariant.*

5. *With probability 1, $A$ has full rank.*

**Assumption 3.** *Suppose $A^{M,N}$ is a sequence of matrices that satisfy Assumptions 2 such that $M, N \to \infty$ with $M/N \to c$. Let $\lambda_1^{M,N}, \ldots, \lambda_{\min(M,N)}^{M,N}$ be the eigenvalues and let $\mu_{M,N} = \sum_i \delta_{\lambda_i^{M,N}}$ be the sum of dirac delta measures for the eigenvalues. Then we shall assume that $\mu_{M,N}$ converges weakly in probability to the Marchenko-Pastur measure $\mu_c$ with shape $c$.*

From here onwards, we shall suppress the superscripts. While such assumptions on the noise may seem restrictive, this encompasses a large family of noise distributions that include Gaussian noise.

**Proposition 1** (Proof in Appendix A). *If $B$ is a random matrix that has full rank with probability one and its entries are independent, have mean 0, have variance $1/M$, and bounded fourth moment, and $P, Q$ are uniformly random orthogonal matrices. Then $A = PBQ$ satisfies Assumptions 2 and 3.*

Note that when we sample matrices as detailed in Assumption 1, we have that $\|X_{trn}\|_F = \|X_{tst}\|_F = 1$. to account for this, let $\theta_{tst}, \theta_{trn} \in \mathbb{R}_+$ be **scalars** that will scale the norms of $X_{trn}, X_{tst}$ so that we can control the SNR of the matrices.

**Assumption 4.** *We assume that $\theta_{tst}$ is fixed and known and that we have control over $\theta_{trn}$.*

Given data $X_{trn}, X_{tst}$ satisfying Assumption 1, noise matrices $A_{trn}, A_{tst}$ satisfying Assumptions 2, 3, and $\theta_{trn}, \theta_{tst}$ that satisfy Assumption 4, then noisy data is given by $Y_{trn} = \theta_{trn} X_{trn} + A_{trn}$ and $Y_{tst} = \theta_{tst} X_{tst} + A_{tst}$.

## 4.3 Assumption about the Model and Training Algorithm

Finally, we shall make assumptions about the denoiser $f$ from Equation 1.

**Assumption 5.** *We shall assume $f$ is a linear model $W$ that is the solution to the following least squares problem.*

$$\min_{\hat{W}} \quad \|\theta_{trn} X_{trn} - \hat{W}(\underbrace{\theta_{trn} X_{trn} + A_{trn}}_{Y_{trn}})\|_F^2. \tag{3}$$

*That is, given data $X_{trn}$ that satisfies Assumption 1, noise matrix $A_{trn}$ that satisfies Assumptions 2, 3, $\theta_{trn}$ that satisfies Assumption 4, and noisy data $Y_{trn} = \theta_{trn} X_{trn} + A_{trn}$, we have that $f(x) = Wx = \theta_{trn} X_{trn} Y_{trn}^{\dagger} x$.*

Here for a matrix $T$, $T^\dagger$ is the Moore-Penrose pseudoinverse. Note here that we are not assuming access to the denoised test data. We rewrite Equation 2 for this denoiser and data generation model.

$$R_{\text{test-error}} := \mathbb{E}_{A_{trn}, A_{tst}} \left[ \frac{\|\theta_{tst}X_{tst} - W(\theta_{tst}X_{tst} + A_{tst})\|_F^2}{N_{tst}} \right]. \tag{4}$$

**Remark 1.** *We analyze this setup instead of the standard Gaussian or Spherical data model since if both our data and noise are isotropic, then the denoising problem can be degenerate. Hence we assume that our data has a low rank.*

**Remark 2.** *While many double descent analyses look at the role of ridge regularization, in this case, since we are looking at the denoising setup, we look at the role of the amount of noise. However, our method can be adapted to include a ridge regularizer.[2] Note that Bishop (1995) shows that adding noise to the input is equivalent to Tikhonov regularization.*

### 4.4 Signal to Noise Ratio (SNR)

A quantity of interest to us will be the SNR, given by $\|X\|_F/\|A\|_F$. Hence, we need to normalize everything by $\|A\|_F$. Due to our assumptions, we have that $\mathbb{E}[\|A\|_F^2] = N$. Hence, for any variables and constants, if it has a hat, then that refers to that variable or constant normalized by $\sqrt{N}$. For example, given $\theta_{trn}, X_{trn}$, and $A_{trn}$, then we have that $\frac{\|\theta_{trn}X_{trn}\|_F}{\|A_{trn}\|_F} = \frac{\theta_{trn}}{\|A_{trn}\|_F} \approx \frac{\theta_{trn}}{\sqrt{N_{trn}}} =: \hat{\theta}_{trn}$.

## 5 Theoretical Results and Consequences

In this section, we analyze the model presented in Section 4. The main theoretical result of the paper is summarized below in Theorem 1. In Theorem 1, for $r = 1$, we exactly characterize the asymptotic generation error.

**Theorem 1.** *Let $r = 1$ and $c = M/N_{trn}$ be fixed. Let $W$ be such that it satisfies Assumption 5 for training data $\theta_{trn}, X_{trn}, Y_{trn}$ that satisfy Assumptions (1-5). Further suppose that $\theta_{trn}$ is $O(\sqrt{N_{trn}})$. Then for test data $\theta_{tst}, X_{tst}, Y_{tst}$ that satisfy Assumptions (1-5) such that $\theta_{tst}$ is $O(\sqrt{N_{tst}})$ the mean squared generalization error (Equation 4) can written as follows. If $c < 1$,*

$$R_{test-error} = \frac{\theta_{tst}^2}{N_{tst}(1+\theta_{trn}^2 c)^2} + o\left(\frac{\theta_{tst}^2}{N_{tst}}\right) + \frac{c^2(\theta_{trn}^2 + \theta_{trn}^4)}{M(1+\theta_{trn}^2 c)^2(1-c)} + o\left(\frac{1}{M}\right) \tag{5}$$

*and if $c > 1$, we have that*

$$R_{test-error} = \frac{\theta_{tst}^2}{N_{tst}(1+\theta_{trn}^2)^2} + o\left(\frac{\theta_{tst}^2}{N_{tst}}\right) + \frac{c\theta_{trn}^2}{M(1+\theta_{trn}^2)(c-1)} + o\left(\frac{1}{M}\right). \tag{6}$$

*The $o\left(\frac{\theta_{tst}^2}{N_{tst}}\right), o\left(\frac{1}{M}\right)$ error terms go to 0 as $N_{trn}, M \to \infty$.*

Theorem 1 is only for rank one data. We do not have the exact generalization error for general low-rank data. However, we can consider the heuristic formulas in Equations 7, 8.[3] Here, the $i$th term in the summation is the rank one formula for the rank one matrix $\sigma_i u_i v_i^T$ corresponding to the $i^{th}$ singular value $\sigma_i$. Here $u_i, v_i$ are the $i^{th}$ singular vectors.

$$\sum_{i=1}^r \frac{(\theta_{tst}\sigma_i^{tst})^2}{N_{tst}(1+(\theta_{trn}\sigma_i^{trn})^2 c)^2} + \frac{c^2((\theta_{trn}\sigma_i^{trn})^2 + (\theta_{trn}\sigma_i^{trn})^4)}{M(1+(\theta_{trn}\sigma_i^{trn})^2 c)^2(1-c)} + o(1) \tag{7}$$

$$\sum_{i=1}^r \frac{(\theta_{tst}\sigma_i^{tst})^2}{N_{tst}(1+(\theta_{trn}\sigma_i^{trn})^2)^2} + \frac{c(\theta_{trn}\sigma_i^{trn})^2}{M(1+(\theta_{trn}\sigma_i^{trn})^2)(c-1)} + o(1). \tag{8}$$

---

[2]See Appendix B for more details.

[3]More details for the heuristic can be found in the Appendix D.1. Here we provide some assumptions under which this is a reasonable formula.

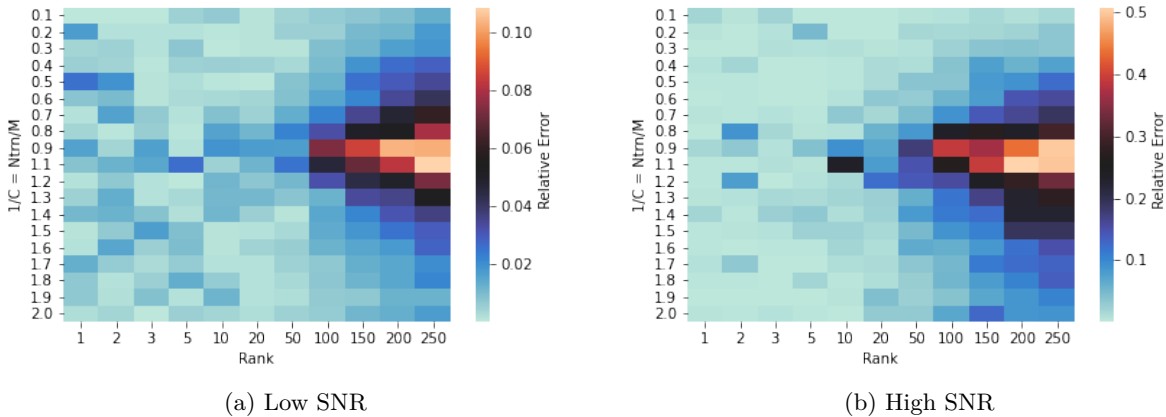

(a) Low SNR        (b) High SNR

Figure 5: Figure showing the accuracy of the heuristic formula for low rank matrices. The figure shows the heatmap of the relative empirical error (|true generalization error - predicted generalization error|/|true generalization error|) when changing $c$ and the rank of the data. Here $M = 2500$ and $c$ is changed by changing $N_{trn}$. We average over ten trials for low SNR, and for high rank, we average over 100.

Before proceeding, we experimentally determine the accuracy of our formula for general rank $r$ data. To do so, we calculate the relative error. That is, if the empirical generalization error is $R_{emp}$ and our theoretical predicition is $R_{theory}$, then we calculate $\dfrac{|R_{emp} - R_{theory}|}{|R_{emp}|}$. Here for low SNR ($\theta_{trn}, \theta_{tst}$ are $O(1)$), we sample $\sigma_i^{trn}, \sigma_i^{tst}$ I.I.D. from the squared standard Gaussian and for high SNR, we multiply the previous by $\sqrt{N_{trn}}, \sqrt{N_{tst}}$ ($\theta_{trn}, \theta_{tst}$ are $\Theta(\sqrt{N_{trn}}), \Theta(\sqrt{N_{tst}})$). As we can see from Figure 5, our formula is accurate for low rank data where we have a relative error of around 0.01. However, we see that the approximation breaks down for higher rank data, especially near $c = 1$.

## 5.1 Data Distributions

While Theorem 1 is only for rank 1 data, the current setup has some general components. In particular, it shows the surprising result that there can be two different types of mismatch between the training data and the test data that do not affect the generalization error.

**Noise Distribution Mismatch.** The first type of mismatch corresponds to the distribution of the noise. Besides the general assumptions on the noise distribution, we note that the distribution for the entries of $A_{trn}$ and the distribution for the entries of $A_{tst}$ need not be the same. Our only restriction is that $A_{trn}$ and $A_{tst}$ satisfy our noise distribution assumptions independently. So, for example, Theorem 1 would apply if we have that the entries of $A_{trn}$ are I.I.D. Gaussian with mean 0 and variance $1/M$ and $A_{tst}$ is sampled by sampling $P, Q$ uniformly from the space of orthogonal matrices and sampling $B$ with I.I.D. entries uniformly on $[-\sqrt{6/M}, \sqrt{6/M}]$ (so that entries have mean 0 and variance $1/M$) and setting $A_{tst} = PBQ$.

**Data Distribution Mismatch.** The next type of mismatch concerns the distribution of $V^{(trn)}$ and $V^{(tst)}$. In particular, we are not assuming that they came from any distribution, just that they satisfy certain assumptions.

In the rank one case, we note that due to our assumptions, we must have that $\sigma_1^{trn} = \sigma_1^{tst} = 1$. Thus, we see that our $i^{th}$ training data point is given by $UV_i^{trn}$ where $U$ is the feature vector and $V_i^{trn}$ is a latent scalar variable. Hence in such a setup, we can imagine the entries of $V^{trn}$ and $V^{tst}$ being drawn independently from some distribution. The only assumption we need to account for is that $\|V^{trn}\| = \|V^{tst}\| = 1$. To account for this, suppose we first sample entries of $\tilde{V}_i^{trn}$ in an I.I.D. manner from some distribution $\mathcal{D}_{trn}$ that has mean 0 and variance 1 and that the entries of $\tilde{V}^{tst}$ are sampled from some distribution $\mathcal{D}_{tst}$ that has mean 0 and variance 1. Then if $N_{trn}, N_{tst}$ are large, then due to the law of large numbers, we have

that with high probability $\frac{1}{N_{trn}}\|\tilde{V}^{trn}\|_F^2 = 1 + o(1) = \frac{1}{N_{tst}}\|\tilde{V}^{(tst)}\|_F^2$. Thus, if we let $V^{trn} = \frac{1}{\|\tilde{V}^{trn}\|_F}\tilde{V}^{trn}$ and $V^{tst} = \frac{1}{\|\tilde{V}^{tst}\|_F}\tilde{V}^{tst}$ with $\theta_{trn} = \hat{\theta}_{trn}\|\tilde{V}^{trn}\|_F$ and $\theta_{tst} = \hat{\theta}_{tst}\|\tilde{V}^{tst}\|_F$ then we see that the $V$s satisfy the general assumptions and with high probability the $\theta$s satisfy the assumptions for Theorem 1.

## 5.2 Insights and Phenomena

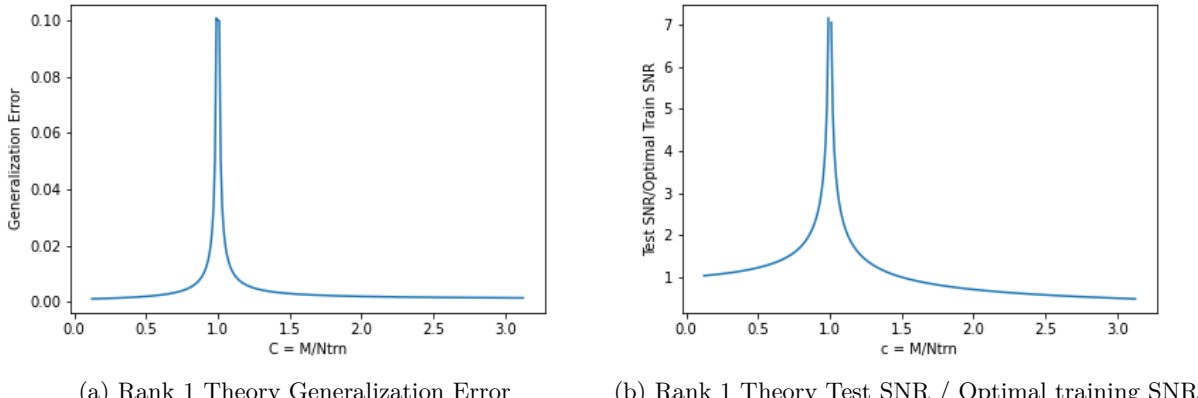

(a) Rank 1 Theory Generalization Error  (b) Rank 1 Theory Test SNR / Optimal training SNR

Figure 6: Plot showing the theoretical double descent curves for the generalization error and the ratio of the test SNR to the optimal training SNR. Here $M = 1000$ and $\theta_{tst} = 1$ and $c$ was changed by changing $N_{trn}$.

Now that we have Theorem 1, we extract a few insights. Specifically, we are interested in insights in the context of the experiments run in Section 3.

### 5.2.1 Optimal Amount of Noise.

If we ignore the error term, we can differentiate the formula to get the following formula for the optimal training SNR. Here $x^+ = \max(0, x)$.

$$\frac{\theta_{opt-trn}^2}{N_{trn}} := \begin{cases} \left(\frac{\theta_{tst}^2}{N_{tst}}\frac{2(1-c)}{2-c} - \frac{c}{M(2-c)}\right)^+ & c < 1 \\ \left(2\frac{\theta_{tst}^2}{N_{tst}}(c-1) - \frac{1}{N_{trn}}\right)^+ & c > 1 \end{cases}. \tag{9}$$

Our theoretical model captures the surprising result that the optimal training SNR and the test SNR are unequal. Moreover, we see that the optimal training distribution depends on $c$. Further, the formula in Equation 9 also describes a double descent curve for $\hat{\theta}_{tst}^2/\hat{\theta}_{opt-trn}^2$ versus $c$ curve as shown in Figure 6b. Thus, we see that our model captures phenomena 2, 3, and 4 from Section 3.

### 5.2.2 Double Descent Curves.

We have already seen that the optimal amount of training noise follows a double decent curve. This double descent is due to the double descent for the generalization error. To understand this phenomenon, we first note that the first term gives the bias of our model in the formula in Theorem 1, and the second term gives the variance. We can see that the variance formulas have a singularity at $c = 1$. Since we have a linear model, $c = 1$ is the interpolation threshold (i.e., the point after which we have 0 training error). Hence, as we approach the interpolation threshold, the model's variance increases, increasing the generalization error. Thus our model captures phenomenon 1. Further, we can see that decreasing $\theta_{trn}$ decreases the model's variance. Since the variance increases near the interpolation threshold, we try to mitigate this by increasing the amount of noise (or reducing $\theta_{trn}$). Hence sample wise double descent for the optimal noise level occurs as a result of trying to reduce the variance of the model. Thus, our model captures the first four phenomena

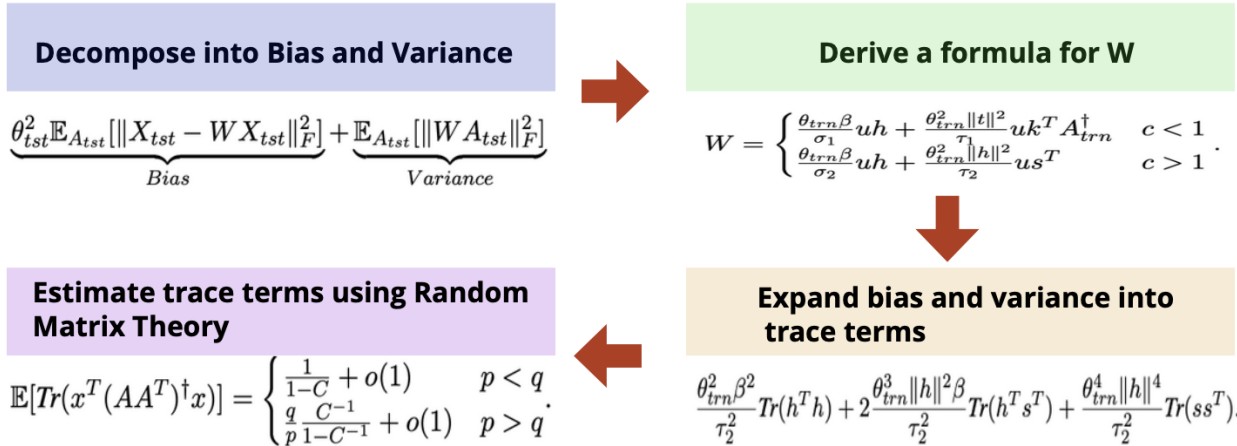

Figure 7: Figure showing the major steps used to derive the formula for the generalization error.

observed in Section 3. Phenomena 5 from Section 3 was that optimally picking the amount of training noise mitigated double descent. However, our theoretical model still has double descent even if we optimally pick the amount of training noise. This is an avenue for future work.

We also compare Theorem 1 to Theorem 1 from Hastie et al. (2022). In Hastie et al. (2022), they assume that they have data $x_i \in \mathbb{R}^M$ from some distribution $\mathcal{D}$, and response $y_i = x_i^T \beta + \xi_i$, where $\beta \in \mathbb{R}^M$ is fixed and $\xi_i \sim \mathcal{N}(0, \sigma^2)$. Then they have the following risk

$$R_X(\hat{\beta}; \beta) = \mathbb{E}_{x_0 \sim \mathcal{D}}[(x_0^T \hat{\beta} - x_0^T \beta)^2 | X_{trn}].$$

This is the conditional excess risk given the training data. Under some assumptions, they show that

$$R_X(\hat{\beta}, \beta) \to \begin{cases} \sigma^2 \frac{c}{1-c} & c < 1 \\ \|\beta\|^2 \left(1 - \frac{1}{c}\right) + \sigma^2 \frac{1}{c-1} & c > 1 \end{cases}.$$

First, we note the similarities between the two. In both cases, we see that the peak is at $c = 1$ and is due to a term of the same order, i.e. both have $(c-1)^{-1}$, and not $(c-1)^{-\alpha}$ for some other $\alpha > 0$.

However, there are differences. First, as detailed in the introduction, they look at the supervised setting, whereas we look at the unsupervised setting. Second, they have input noise, whereas we have output noise. Third, they have zero bias in the under-parameterized regime. However, we have a non-zero bias term in both the over and under-parameterized regimes.

### 5.2.3 Noise as a Regularizer.

Finally, we see that noise level explicitly regularizes $\|W\|_F$. Specifically, from Lemmas 1 and 3, the second term in Theorem 1 corresponds to $\|W\|_F$. The formula shows that increasing the amount of noise, which corresponds to decreasing $\theta_{trn}$, decreases $\|W\|_F$.

## 6 Proof of Theorem 1

We prove Theorem 1 via the steps shown in Figure 7. The proofs for all of the lemmas have been moved to Appendix C. Here we present a proof sketch that details the high-level ideas.

### 6.1 Step 1: Decompose the error into bias and variance terms.

First, we decompose the error. Since we are not in the supervised learning setup, we do not have standard definitions of bias/variance. However, we will call the following terms the bias/variance of the model.

**Lemma 1.** *If $A_{tst}$ has mean 0 entries and $A_{tst}$ is independent of $X_{tst}$ and $W$, then*

$$\mathbb{E}_{A_{tst}}[\|\theta_{tst}X_{tst} - WY_{tst}\|_F^2] = \underbrace{\theta_{tst}^2\mathbb{E}_{A_{tst}}[\|X_{tst} - WX_{tst}\|_F^2]}_{Bias} + \underbrace{\mathbb{E}_{A_{tst}}[\|WA_{tst}\|_F^2]}_{Variance}. \tag{10}$$

### 6.2 Step 2: Formula for $W$

In our current setup, $W$ is the solution to a least-squares problem. Hence $W = \theta_{trn}X_{trn}Y_{trn}^\dagger$. Expanding this out, we get the following formula for $W$. Let $u$ be the left singular vector and $v_{trn}, v_{tst}$ the right singular vectors. Let $h = v_{trn}^T A_{trn}^\dagger$, $k = A_{trn}^\dagger u$, $s = (I - A_{trn}A_{trn}^\dagger)u$, $t = v_{trn}(I - A_{trn}^\dagger A_{trn})$, $\beta = 1 + \theta_{trn}v_{trn}^T A_{trn}^\dagger u$, $\tau_1 = \theta_{trn}^2\|t\|^2\|k\|^2 + \beta^2$, and $\tau_2 = \theta_{trn}^2\|s\|^2\|h\|^2 + \beta^2$.

**Proposition 2.** *If $\beta \neq 0$ and $A_{trn}$ has full rank then* $W = \begin{cases} \frac{\theta_{trn}\beta}{\tau_1}uh + \frac{\theta_{trn}^2\|t\|^2}{\tau_1}uk^T A_{trn}^\dagger & c < 1 \\ \frac{\theta_{trn}\beta}{\tau_2}uh + \frac{\theta_{trn}^2\|h\|^2}{\tau_2}us^T & c > 1 \end{cases}$.

For Gaussian noise, $A_{trn}$ has full rank with probability one, and $\beta$ is a random variable whose expected value equals 1, and the distribution is highly concentrated. Thus, Proposition 2 applies when $A_{trn}$ is isotropic Gaussian noise. Here we restricted ourselves to rank 1, as using Meyer (1973), we can expand formulas of the form $(A + xy^T)^\dagger$ where $x, y$ are vectors. For the higher rank case, we apply the formula iteratively. This is the main difficulty of the method. Previous work on deriving asymptotics for the generalization error had noise on the output. Hence would take the pseudoinverse of a matrix that only depended on the data. However, in our case, we are taking the pseudoinverse of a matrix that depends on the noise.

### 6.3 Step 3: Decompose the terms into a sum of various trace terms.

For the bias and variance terms, we have the following two Lemmas.

**Lemma 2.** *If $W$ is the solution to Equation 3, then* $X_{tst} - WX_{tst} = \begin{cases} \frac{\beta}{\tau_1}X_{tst} & if\ c < 1 \\ \frac{\beta}{\tau_2}X_{tst} & if\ c > 1 \end{cases}$.

**Lemma 3.** *If the entries of $A_{tst}$ are independent with mean 0, and variance $1/M$, then we have that* $\mathbb{E}_{A_{tst}}[\|WA_{tst}\|^2] = \frac{N_{tst}}{M}\|W\|^2$.

Note that this did not need assumptions on $W$ or $X_{tst}$. All that was needed were the assumptions on $A_{tst}$. Thus, this holds more generally. This decomposition also follows from Bishop (1995). In light of Lemmas 1, 2, 3, and the fact that $\|X_{tst}\|_F^2 = \theta_{tst}^2$, we see that the expected mean squared generalization error is given by,

$$\mathbb{E}_{A_{tst}}\left[\frac{\|\theta_{tst}X_{tst} - WY_{tst}\|_F^2}{N_{tst}}\right] = \frac{1}{N_{tst}}\frac{\beta^2}{\tau_i^2}\theta_{tst}^2 + \frac{1}{M}\|W\|_F^2,$$

where $\tau_i$ depends on whether $c < 1$ or $c > 1$. Finally, let us look at the $\|W\|$ term.

**Lemma 4.** *If $\beta \neq 0$ and $A_{trn}$ has full rank, then we have that if $c < 1$,*

$$\|W\|_F^2 = \frac{\theta_{trn}^2\beta^2}{\tau_1^2}Tr(h^T h) + 2\frac{\theta_{trn}^3\|t\|^2\beta}{\tau_1^2}Tr(h^T k^T A_{trn}^\dagger) + \frac{\theta_{trn}^4\|t\|^4}{\tau_1^2}Tr((A_{trn}^\dagger)^T kk^T A_{trn}^\dagger)$$

*and if $c > 1$, then we have that*

$$\|W\|_F^2 = \frac{\theta_{trn}^2\beta^2}{\tau_2^2}Tr(h^T h) + 2\frac{\theta_{trn}^3\|h\|^2\beta}{\tau_2^2}Tr(h^T s^T) + \frac{\theta_{trn}^4\|h\|^4}{\tau_2^2}Tr(ss^T).$$

### 6.4 Step 4: Estimate using random matrix theory.

While the formula given by Lemmas 1, 3, and 4 is correct, we need a simpler formula to analyze the situation. Using ideas from random matrix theory, we can simplify the expression for $\|W\|_F^2$. To do so, we first need to prove Lemmas 5 and 6. The main idea behind Lemmas 5 and 6 is that due to the rotational invariance of $A_{trn}$,

the expectation of the trace of products of various matrices derived from $A_{trn}$ is determined by the expected value of some function $\chi$ of the eigenvalues of $A_{trn}$. However, instead of directly computing this expected value, we note that for any matrix $A$ that satisfies the noise assumptions, if we let $M, N \to \infty$, with $M/N \to c$, then the eigenvalue distribution converges to the Marchenko - Pastur distribution (Marcenko & Pastur, 1967; Götze & Tikhomirov, 2011; 2003; 2004; 2005; Bai et al., 2003). Götze & Tikhomirov (2004) showed that the distribution of the eigenvalues converged almost surely with a rate of at least $O(N^{-1/2+\epsilon})$ for any $\epsilon > 0$. Thus, we can use the expected value of the $\chi(\lambda)$ for $\lambda$ sampled from the Marchenko - Pastur distribution as an approximation.

**Lemma 5.** *Suppose $A$ is an $p$ by $q$ matrix such that the entries of $A$ are independent and have mean 0, variance $1/q$, and bounded fourth moment. Let $W_p = AA^T$ and let $W_q = A^T A$. Let $C = p/q$. Suppose $\lambda_p, \lambda_q$ are a random eigenvalue of $W_p, W_q$. Then*

1. If $p < q$, then $\mathbb{E}\left[\frac{1}{\lambda_p}\right] = \frac{1}{1-C} + o(1)$.

2. If $p < q$, then $\mathbb{E}\left[\frac{1}{\lambda_p^2}\right] = \frac{1}{(1-C)^3} + o(1)$.

3. If $p < q$, then $\mathbb{E}\left[\frac{1}{\lambda_p^3}\right] = \frac{1}{(1-C)^5} + o(1)$.

4. If $p < q$, then $\mathbb{E}\left[\frac{1}{\lambda_p^4}\right] = \frac{C^2 + \frac{22}{6}C + 1}{(1-C)^7} + o(1)$.

5. If $p > q$, then $\mathbb{E}\left[\frac{1}{\lambda_q}\right] = \frac{C^{-1}}{1-C^{-1}} + o(1)$.

6. If $p > q$, then $\mathbb{E}\left[\frac{1}{\lambda_q^2}\right] = \frac{C^{-2}}{(1-C^{-1})^3} + o(1)$.

7. If $p > q$, then $\mathbb{E}\left[\frac{1}{\lambda_q^3}\right] = \frac{C^{-3}(1+C^{-1})}{(1-C^{-1})^5} + o(1)$.

8. If $p > q$, then $\mathbb{E}\left[\frac{1}{\lambda_q^4}\right] = \frac{C^{-4}(C^{-2}+\frac{22}{6}C^{-1}+1)}{(1-C^{-1})^7} + o(1)$.

**Lemma 6.** *Suppose $A$ is an $p$ by $q$ matrix such that the entries of $A$ are independent and have mean 0, variance $1/q$, and bounded fourth moment. Let $C = p/q$ and let $x, y$ be unit vectors in $p$, then*

1. $\mathbb{E}[Tr(x^T(AA^T)^\dagger x)] = \begin{cases} \frac{1}{1-C} + o(1) & p < q \\ \frac{q}{p}\frac{C^{-1}}{1-C^{-1}} + o(1) & p > q \end{cases}$.

2. $\mathbb{E}[Tr(x^T(AA^T)^\dagger (AA^T)^\dagger x)] = \begin{cases} \frac{1}{(1-C)^3} + o(1) & p < q \\ \frac{q}{p}\frac{C^{-2}}{(1-C^{-1})^3} + o(1) & p > q \end{cases}$.

Using these technical lemmas, we can now deal with all of the terms in the expressions in Lemma 4.

**Lemma 7.** *If $A_{trn}$ satisfies the noise assumptions, then we have that*

1. $\mathbb{E}[\beta/\theta_{trn}] = 1/\theta_{trn} + o(1)$ and $\text{Var}(\beta/\theta_{trn}) = \frac{c}{(\max(M,N_{trn})|1-c|))} + o(1)$.

2. If $c < 1$, then $\mathbb{E}[\|h\|^2] = \frac{c^2}{1-c} + o(1)$ and $\text{Var}(\|h\|^2) = \frac{c^3(2+c)}{N_{trn}(1-c)^3} + o(1)$.

3. If $c > 1$, then $\mathbb{E}[\|h\|^2] = \frac{c}{c-1} + o(1)$ and $\text{Var}(\|h\|^2) = \frac{c^2(2c-1)}{N_{trn}(c-1)^3} + o(1)$.

4. $\mathbb{E}[\|k\|^2] = \frac{c}{1-c} + o(1)$ and $\text{Var}(\|k\|^2) = \frac{c^2(2+c)}{M(1-c)^3} + o(1)$.

5. $\mathbb{E}[\|s\|^2] = \frac{c-1}{c} + o(1)$ and $\text{Var}(\|s\|^2) = 2\frac{1}{Mc} + o(1)$.

6. $\mathbb{E}[\|t\|^2] = 1 - c + o(1)$, $\text{Var}(\|t\|^2) = 2\frac{c}{N_{trn}} + o(1)$.

**Lemma 8.** *Under the noise assumptions, we have that $\mathbb{E}[Tr(h^T k^T A_{trn}^\dagger)] = 0$ and $\text{Var}(Tr(h^T k^T A_{trn}^\dagger)) = \chi_3(c)/N_{trn}$, where $\chi_3(c) = \mathbb{E}[1/\lambda^3]$, $\lambda$ is an eigenvalue for $AA^T$ and $A$ is as in Lemma 6.*

**Lemma 9.** *Under the noise assumptions, we have that*

$$Tr((A_{trn}^\dagger)^T kk^T A_{trn}^\dagger) = \frac{c^2}{(1-c)^3} + o(1), \quad \text{Var}(Tr((A_{trn}^\dagger)^T kk^T A_{trn}^\dagger)) = \frac{3}{M}\chi_4(c) - \frac{1}{M}\frac{c^4}{(1-c)^6}$$

*where $\chi_4(c) = \mathbb{E}[1/\lambda^4]$, $\lambda$ is an eigenvalue for $AA^T$ and $A$ is as in Lemma 6.*

**Lemma 10.** *Under the same assumptions as Proposition 2, we have that $Tr(h^T s^T) = 0$.*

Lemmas 7, 8, 9, and 10 tell us that all of the terms are highly concentrated. Thus, even though such terms may not be uncorrelated, we can use the fact that $|\mathbb{E}[XY] - \mathbb{E}[X]\mathbb{E}[Y]| < \sqrt{\text{Var}(X)\text{Var}(Y)}$, to treat the terms as

if they are uncorrelated. Since these variances have now been shown to be $o(1)$, we have that for each of these terms $\mathbb{E}[XY] = \mathbb{E}[X]\mathbb{E}[Y]+o(1)$. For example, since $\tau_1 = \beta^2+\theta_{trn}^2\|t\|^2\|k\|^2+o(1)$, using Lemmas 1, 4, and 6, we have that $\mathbb{E}[\tau_1] = 1+\theta_{trn}^2 c+o(1)$. Similarly, $\mathbb{E}[\tau_2] = 1+\theta_{trn}^2+o(1)$. Finally, using these lemmas, we can simplify the expressions in Lemma 4 to get the formulas for the expected generalization error shown in Equations 5 and 6.

## 7 Conclusion

In this paper, we switch focus from a supervised setup to an unsupervised setup. Specifically, we look at the problem of denoising data. We empirically show five interesting phenomena in our given setup. First, we see sample-wise double descent for the generalization error for denoising feedforward neural networks. Second, we see that, under certain circumstances, the optimal denoising error does not occur when the training data SNR is equal to the test data SNR. Third, we see that the optimal ratio depends on the number of data points. Fourth, we see that curve also has sample-wise double descent, and fifth, picking the correct training noise level mitigates sample-wise double descent of the generalization error. To provide theoretical analysis for this model, we look at a theoretical model where our data has a low rank. Here we derive the exact asymptotics for the generalization error for rank 1 data and a general noise model. Our analysis demonstrates that this simple model captures most of the phenomena seen empirically.

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

In this section we present all of the proofs for the results in the main text. Here we present the proofs in the same order they appear in the text.

## A    Noise Assumptions

**Proposition 1.** *If $B$ is a random matrix that has full rank with probability 1 and its entries are independent, have mean 0, and have variance $1/M$ and $P, Q$ are uniformly random orthogonal matrices. Then $A = PBQ$ satisfies all of our noise assumptions.*

*Proof.* Since $P, Q$ are a uniformly random orthogonal matrices, and $A = PBQ$, then it is clear that $A$ is rotationally bi-invariant and has full rank.

Since each entry of $B$ has mean 0 and each entry of $A$ is a linear combination of entries of $B$ where the coefficients (i.e., the entries from $P, Q$ are independent of $B$), we have that each entry of $B$ have mean 0. Due to the orthogonal nature of $P, Q$, we have the variance for an entry of $A$ is the same as the variance of entry in $B$.

Thus, the only thing left to prove is that the entries of $A$ are uncorrelated. To do this, we note that

$$a_{ij} = \sum_{k=1}^{N} \sum_{l=1}^{M} p_{il} b_{lk} q_{kj}.$$

Consider two entries $a_{i_1 j_1}$ and $a_{i_2 j_2}$. Then we have that

$$
\begin{aligned}
\mathbb{E}[a_{i_1 j_1} a_{i_2 j_2}] &= \mathbb{E}\left[ \left( \sum_{k=1}^{N} \sum_{l=1}^{M} p_{i_1 l} b_{lk} q_{k j_1} \right) \left( \sum_{k=1}^{N} \sum_{l=1}^{M} p_{i_2 l} b_{lk} q_{k j_2} \right) \right] \\
&= \sum_{k=1}^{N} \sum_{l=1}^{M} \mathbb{E}[p_{i_1 l} p_{i_2 l}] \mathbb{E}[b_{lk}^2] \mathbb{E}[q_{k j_1} q_{k j_2}] \\
&= \frac{1}{M} \mathbb{E}\left[ \sum_{l=1}^{M} p_{i_1 l} p_{i_2 l} \right] \mathbb{E}\left[ \sum_{k=1}^{N} q_{k j_1} q_{k j_2} \right].
\end{aligned}
$$

The second inequality follows from the fact that $P, Q, B$ are independent from each other, and that fact that the entries of $B$ are independent and have mean 0. Hence the cross terms have expectation 0. If we have that $i_1 = i_2$ and $j_1 \neq j_2$, then we have that since $Q$ is an orthogonal matrix

$$\sum_{k=1}^{N} \mathbb{E}[q_{k j_1} q_{k j_2}] = \mathbb{E}\left[ \sum_{k=1}^{N} q_{k j_1} q_{k j_2} \right] = 0.$$

Thus, the entries are uncorrelated. Similarly when $i_1 \neq i_2$ since $P$ is orthogonal matrix, we get that the entries are uncorrelated. $\qquad\square$

**Convergence to Marchenko-Pastur.** If we strengthened the uncorrelated condition, to the entries being independent. Then due to the mean and variance assumptions (along with an assumption that the fourth moment is bounded), we would have convergence to Marchenko-Pastur distribution. However, the independence along with the bi-invariance would then force our noise distribution to be i.i.d. Gaussian.

In general however, with relaxed assumption of the entries only being uncorrelated, convergence is not known. However, in our case, we have a much simpler proof for matrices formed by Proposition 1. In our case, the noise matrices $B$ satisfy the standard assumptions for convergence. We then multiply $B$ by orthogonal matrices that are independent to $B$. Hence this has no effect on the eigenvalue distribution. Thus, the eigenvalues distribution for these matrices also converge to the Marchenko-Pastur distribution.

## B  Ridge Regularization

Here we are now interested in minimizing

$$\|\theta_{trn}X_{trn} - W(\theta_{trn}X_{trn} + A_{trn})\|_F^2 + \mu^2\|W\|_F^2.$$

This problem is equivalent to minimizing

$$\left\|\theta_{trn}\begin{bmatrix}X_{trn} & 0\end{bmatrix} - W\left(\theta_{trn}\begin{bmatrix}X_{trn} & 0\end{bmatrix} + \begin{bmatrix}A_{trn} & \lambda I\end{bmatrix}\right)\right\|_F^2.$$

Thus using $\tilde{A}_{trn} = \begin{bmatrix}A_{trn} & \lambda I\end{bmatrix}$. This is the same problem as before but with different assumptions on the noise matrix. Note that Lemma 1 still applies. As does Proposition 2 but with $\tilde{A}_{trn}$ instead of $A_{trn}$ and $v_{trn}$ has appended zeros. Hence the rest of the proof is similar and we need to look at eigenvalues of $\tilde{A}_{trn}^T\tilde{A}_{trn}$ instead of $A_{trn}^T A_{trn}$. Here we note that

$$\tilde{A}_{trn}^T\tilde{A}_{trn} = A_{trn}^T A_{trn} + \mu^2 I.$$

Thus we have that the eigenvalues are shifted by $\mu^2$. We need to explicitly deal with this during calculation and will need to modify Lemma 5, and need to adjust our calculations accordingly.

## C  Proofs

Due to our data generation assumptions that $\|\Sigma_{trn}\|_F = \|\Sigma_{tst}\|_F = 1$ for rank 1 data, we have that $\sigma_1^{trn} = \sigma_1^{tst} = 1$.

### C.1  Step 1: Decompose into bias and Varaince

**Lemma 1.** *If $A_{tst}$ has mean 0 entries and $A_{tst}$ is independent of $X_{tst}$ and $W$, then*

$$\mathbb{E}_{A_{tst}}[\|\theta_{tst}X_{tst} - WY_{tst}\|_F^2] = \underbrace{\theta_{tst}^2\mathbb{E}_{A_{tst}}[\|X_{tst} - WX_{tst}\|_F^2]}_{Bias} + \underbrace{\mathbb{E}_{A_{tst}}[\|WA_{tst}\|_F^2]}_{Variance}.$$

*Proof.* Using the fact that for any two matrices $\|G - H\|_F^2 = \|G\|_F^2 + \|H\|_F^2 - 2\text{Tr}(G^T H)$, we get that

$$
\begin{aligned}
\|\theta_{tst}X_{tst} - WY_{tst}\|^2 &= \|\theta_{tst}X_{tst} - W\theta_{tst}X_{tst} - WA_{tst}\|_F^2 \\
&= \theta_{tst}^2\|X_{tst} - WX_{tst}\|_F^2 + \|WA_{tst}\|^2 - 2\text{Tr}((\theta_{tst}X_{tst} - W\theta_{tst}X_{tst})^T WA_{tst}).
\end{aligned}
$$

Then since the trace is linear, and $X_{tst}, W$ are independent of $A_{tst}$, and $A_{tst}$ has mean 0 entries, we see that

$$\mathbb{E}_{A_{tst}}[\text{Tr}((\theta_{tst}X_{tst} - W\theta_{tst}X_{tst})^T WA_{tst})] = 0.$$

Thus, we have the needed result. □

### C.2  Step 2: Formula for $W_{opt}$

**Proposition 2.** *Let $h = v_{trn}^T A_{trn}^\dagger$, $k = A_{trn}^\dagger u$, $s = (I - A_{trn}A_{trn}^\dagger)u$, $t = v_{trn}(I - A_{trn}^\dagger A_{trn})$, $\beta = 1 + \theta_{trn}v_{trn}^T A_{trn}^\dagger u$, $\tau_1 = \theta_{trn}^2\|t\|^2\|k\|^2 + \beta^2$, and $\tau_2 = \theta_{trn}^2\|s\|^2\|h\|^2 + \beta^2$. If $\beta \neq 0$ and $A_{trn}$ has full rank then*

$$W_{opt} = \begin{cases} \frac{\theta_{trn}\beta}{\tau_1}uh + \frac{\theta_{trn}^2\|t\|^2}{\tau_1}uk^T A_{trn}^\dagger & c < 1 \\ \frac{\theta_{trn}\beta}{\tau_2}uh + \frac{\theta_{trn}^2\|h\|^2}{\tau_2}us^T & c > 1 \end{cases}.$$

*Proof.* Let us first proof the case when $c > 1$. Here we know that $u$ is arbitrary. Here we have that $A_{trn}$ has full rank. Thus, since $c > 1$, we have that $M > N_{trn}$, thus $A_{trn}$ has rank $N_{trn}$. Thus, the rows of $A_{trn}$ span

the whole space. Thus, $v_{trn}$ lives in the range of $A_{trn}^T$. Finally, since $\beta \neq 0$, we want Theorem 5 from Meyer (1973).

Here let us further define

$$p_2 = -\frac{\theta_{trn}^2 \|s\|^2}{\beta} A_{trn}^\dagger h^T - \theta_{trn}k \text{ and } q_2^T = -\frac{\theta_{trn}\|h\|^2}{\beta}s^T - h$$

and finally $\tau_2 = \theta_{trn}^2 \|s\|^2 \|h\|^2 + \beta^2$. Then we have from Meyer (1973) that

$$(A_{trn} + \theta_{trn}uv_{trn}^T)^\dagger = A_{trn}^\dagger + \frac{\theta_{trn}}{\beta}A_{trn}^\dagger h^T s^T - \frac{\beta}{\tau_2}p_2 q_2^T$$

In our case, we only care about $\theta_{trn}uv_{trn}^T(A_{trn} + \theta_{trn}uv_{trn}^T)^\dagger$. Thus let us multiply this through and see what we get.

$$\theta_{trn}uv_{trn}^T(A_{trn} + \theta_{trn}uv_{trn}^T)^\dagger = \theta_{trn}uv_{trn}^T(A_{trn}^\dagger + \frac{\theta_{trn}}{\beta}A^\dagger h^T s^T - \frac{\beta}{\tau_2}p_2 q_2^T)$$

$$= \theta_{trn}uh + \frac{\theta_{trn}^2\|h\|^2}{\beta}us^T + \frac{\theta_{trn}\beta}{\tau_2}uv_{trn}^T\left(\frac{\theta_{trn}^2\|s\|^2}{\beta}A_{trn}^\dagger h^T + \theta_{trn}k\right)q_2^T$$

$$= \theta_{trn}uh + \frac{\theta_{trn}^2\|h\|^2}{\beta}us^T + \frac{\theta_{trn}^3\|s\|^2\|h\|^2}{\tau_2}uq_2^T + \frac{\theta_{trn}^2\beta}{\tau_2}uhuq_2^T$$

Then we have that

$$\frac{\theta_{trn}^3\|s\|^2\|h\|^2}{\tau_2}cq_2^T = -\frac{\theta_{trn}^4\|s\|^2\|h\|^4}{\tau_2\beta}us^T - \frac{\theta_{trn}^3\|s\|^2\|h\|^2}{\tau_2}uh \tag{11}$$

and

$$\frac{\theta_{trn}^2\beta}{\tau_2}uhuq_2^T = -\frac{\theta_{trn}^3\|h\|^2}{\tau_2}uhus^T - \frac{\theta_{trn}^2\beta}{\tau_2}uhuh. \tag{12}$$

Using that $\beta - 1 = \theta_{trn}v_{trn}^T A_{trn}^\dagger u = \theta_{trn}hu$, we get that

$$\frac{\theta_{trn}^2\beta}{\tau_2}uhuq_2^T = -\frac{\theta_{trn}^2\|h\|^2(\beta-1)}{\tau_2}us^T - \frac{\theta_{trn}\beta(\beta-1)}{\tau_2}uh. \tag{13}$$

Substituting back in and collecting like terms we get that

$$\theta_{trn}uv_{trn}^T(A_{trn} + \theta_{trn}uv_{trn}^T)^\dagger = \theta_{trn}u\left(1 - \frac{\theta_{trn}^2\|s\|^2\|h\|^2}{\tau_2} - \frac{\beta(\beta-1)}{\tau_2}\right)h +$$

$$\theta_{trn}^2u\left(\frac{\|h\|^2}{\beta} - \frac{\theta_{trn}^2\|s\|^2\|h\|^4}{\tau_2\beta} - \frac{\|h\|^2(\beta-1)}{\tau_2}\right)s^T$$

We can then simplify the constants as follows.

$$1 - \frac{\theta_{trn}^2\|s\|^2\|h\|^2}{\tau_2} - \frac{\beta(\beta-1)}{\tau_2} = \frac{\tau_2 - \theta_{trn}^2\|s\|^2\|h\|^2 - \beta^2 + \beta}{\tau_2} = \frac{\beta}{\tau_2}$$

and

$$\frac{\|h\|^2}{\beta} - \frac{\theta_{trn}^2\|s\|^2\|h\|^4}{\tau_2\beta} - \frac{\|h\|^2(\beta-1)}{\tau_2} = \frac{\|h\|^2(\tau_2 - \theta_{trn}^2\|s\|^2\|h\|^2 - \beta(\beta-1))}{\beta\tau_2} = \frac{\|h\|^2\beta}{\beta\tau_2} = \frac{\|h\|^2}{\tau_2}.$$

This gives us the result for $c > 1$.

If $c < 1$, then we have that $M < N_{trn}$. Thus, the rank of $A_{trn}$ is $M$ the range of $A_{trn}$ is the whole space. Thus, $u$ lives in the range of $A_{trn}$. In this case, we then want Theorem 3 from Meyer (1973). In this case, we define

$$p_1 = -\frac{\theta_{trn}^2 \|k\|^2}{\beta} t^T - k \text{ and } q_1^T = -\frac{\theta_{trn} \|t\|^2}{\beta} k^T A_{trn}^\dagger - h.$$

Then in this case, we have that

$$(A_{trn} + \theta_{trn} u v_{trn}^T)^\dagger = A_{trn}^\dagger + \frac{\theta_{trn}}{\beta} t^T k^T A_{trn}^\dagger - \frac{\beta}{\tau_1} p_1 q_1^T.$$

Then we simplify the equation as we did before! □

### C.3   Step 3: Expand into trace terms

**Lemma 3.** *If the entries of $A_{tst}$ are independent with mean 0, and variance $1/M$, then we have that* $\mathbb{E}_{A_{tst}}[\|W A_{tst}\|^2] = \frac{N_{tst}}{M} \|W\|^2$.

*Proof.* To see this, we note if we look at $A_{tst} A_{tst}^T$, then this is a $M$ by $M$, for which the expected value of the off diagonal entries is equal to 0, while the expected value of each diagonal entry is $N_{tst}/M$. That is, $\mathbb{E}_{A_{tst}}[A_{tst} A_{tst}^T] = \frac{N_{tst}}{M} I_M$.

Then note that

$$\|W A_{tst}\|^2 = \text{Tr}(A_{tst}^T W^T W A_{tst}) = \text{Tr}(W^T W A_{tst} A_{tst}^T) = \text{Tr}(W^T W A_{tst} A_{tst}^T).$$

Using the fact that the trace is linear again, we see that

$$\mathbb{E}_{A_{tst}}[\text{Tr}(W^T W A_{tst} A_{tst}^T)] = \text{Tr}(W^T W \mathbb{E}_{A_{tst}}[A_{tst} A_{tst}^T]) = \frac{N_{tst}}{M} \text{Tr}(W^T W) = \frac{N_{tst}}{M} \|W\|_F^2.$$

□

**Lemma 2.** *If $W$ is the solution to Equation 3, then*

$$X_{tst} - W X_{tst} = \begin{cases} \frac{\beta}{\tau_1} X_{tst} & if\ c < 1 \\ \frac{\beta}{\tau_2} X_{tst} & if\ c > 1 \end{cases}.$$

*Proof.* To see this, we have the following calculation for when $N_{trn} > M$.

$$X_{tst} - W X_{tst} = X_{tst} - \frac{\theta_{trn} \beta}{\tau_1} u h u v_{tst}^T - \frac{\theta_{trn}^2 \|t\|^2}{\tau_1} u k^T A_{trn}^\dagger u v_{tst}^T$$

$$= X_{tst} - \frac{\theta_{trn} \beta}{\tau_1} u v_{trn}^T A_{trn}^\dagger u v_{tst}^T - \frac{\theta_{trn}^2 \|t\|^2}{\tau_1} u k^T A_{trn}^\dagger u v_{tst}^T.$$

First, we note that $\beta = 1 + \theta_{trn} v_{trn}^T A_{trn}^\dagger u$. Thus, we have that $\theta v_{trn}^T A_{trn}^\dagger u = \beta - 1$. Thus, substituting this into the second term, we get that

$$X_{tst} - W X_{tst} = X_{tst} - \frac{\beta(\beta - 1)}{\tau_1} u v_{tst}^T - \frac{\theta_{trn}^2 \|t\|^2}{\tau_1} u k^T A_{trn}^\dagger u v_{tst}^T.$$

For the third term, we note that $k = A_{trn}^\dagger u$. Thus, we have that $k^T A_{trn}^\dagger u = k^T k = \|k\|^2$. Substituting this into the expression, we get that

$$X_{tst} - W X_{tst} = X_{tst} - \frac{\beta(\beta - 1)}{\tau_1} u v_{tst}^T - \frac{\theta_{trn}^2 \|t\|^2 \|k\|^2}{\tau_1} u v_{tst}^T.$$

Noting that $X_{tst} = uv_{tst}^T$, we get that

$$X_{tst} - WX_{tst} = X_{tst}\left(1 - \frac{\beta(\beta-1)}{\tau_1} - \frac{\theta_{trn}^2\|t\|^2\|k\|^2}{\tau_1}\right).$$

To simplify the constants, we note that $\tau_1 = \theta_{trn}^2\|t\|^2\|k\|^2 + \beta^2$. Thus, we get that

$$\frac{\tau_1 + \beta - \beta^2 - \theta_{trn}^2\|t\|^2\|k\|^2}{\tau_1} = \frac{\beta}{\tau_1}.$$

For the case when $N_{trn} < M$, we note that the first term of $W$ is the same (modulo replacing $\tau_1$ for $\tau_2$) as it is for the case when $c > 1$. Thus, we just need to deal with the last term. Here we see that the last term is

$$\frac{\theta_{trn}^2\theta_{tst}\|h\|^2}{\tau_2}us^Tuv_{tst}^T.$$

Here we note that $s = (I - A_{trn}A_{trn}^\dagger)u$. Thus, in particular, $s$ is the projection of $u$ onto the kernel of $A_{trn}^T$. Thus, we have that $u = s + \hat{s}$, where $s \perp \hat{s}$. This then tells us that $s^Tu = \|s\|^2$. Thus, for this term, we get that it is equal to

$$\frac{\theta^2\|h\|^2\|s\|^2}{\tau_2}X_{tst}.$$

For this term we note that $\tau_2 = \beta^2 + \theta_{trn}^2\|h\|^2\|u\|^2$. Thus, doing the same simplification as before, we see that for the case when $N_{trn} < M$, we have that

$$X_{tst} - WX_{tst} = \frac{\beta}{\tau_2}X_{tst}.$$

$\square$

In light of Lemma 2 and the fact that $\|\theta_{tst}X_{tst}\|_F^2 = \theta_{tst}^2$. We see that if we look at the expected MSE, we have that,

$$\mathbb{E}_{A_{tst}}\left[\frac{\|\theta_{tst}X_{tst} - W(\theta_{tst}X_{tst} + A_{tst})\|}{N_{tst}}\right] = \frac{\beta}{N_{tst}\tau_i}\theta_{tst}^2 + \frac{1}{M}\|W\|_F^2,$$

where $\tau_i$ depends on whether $c < 1$ or $c > 1$.

Finally, let us look at the $\|W\|$ term.

**Lemma 4.** *If $\beta \neq 0$ and $A_{trn}$ has full rank, then we have that if $c < 1$,*

$$\|W\|_F^2 = \frac{\theta_{trn}^2\beta^2}{\tau_1^2}\,Tr(h^Th) + 2\frac{\theta_{trn}^3\|t\|^2\beta}{\tau_1^2}\,Tr(h^Tk^TA_{trn}^\dagger) + \frac{\theta_{trn}^4\|t\|^4}{\tau_1^2}\,Tr((A_{trn}^\dagger)^Tkk^TA_{trn}^\dagger)$$

*and if $c > 1$, then we have that*

$$\|W\|_F^2 = \frac{\theta_{trn}^2\beta^2}{\tau_2^2}\,Tr(h^Th) + 2\frac{\theta_{trn}^3\|h\|^2\beta}{\tau_2^2}\,Tr(h^Ts^T) + \frac{\theta_{trn}^4\|h\|^4}{\tau_2^2}\,Tr(ss^T).$$

*Proof.* To deal with the term $\mathrm{Tr}(W^TW)$ we are again going to have to look at whether $N_{trn}$ is bigger than or smaller than $M$. First, let us start by looking at the case when $N_{trn} > M$. Here we have that

$$\|W\|_F^2 = \mathrm{Tr}(W^TW)$$
$$= \mathrm{Tr}\left(\left(\frac{\theta_{trn}\beta}{\tau_1}uh + \frac{\theta_{trn}^2\|t\|^2}{\tau_1}uk^TA_{trn}^\dagger\right)^T\left(\frac{\theta_{trn}\beta}{\tau_1}uh + \frac{\theta_{trn}^2\|t\|^2}{\tau_1}uk^TA_{trn}^\dagger\right)\right)$$
$$= \frac{\theta_{trn}^2\beta^2}{\tau_1^2}\mathrm{Tr}(h^Tu^Tuh) + 2\frac{\theta_{trn}^3\|t\|^2\beta}{\tau_1^2}\mathrm{Tr}(h^Tu^Tuk^TA_{trn}^\dagger) + \frac{\theta_{trn}^4\|t\|^4}{\tau_1^2}\mathrm{Tr}((A_{trn}^\dagger)^Tku^Tuk^TA_{trn}^\dagger)$$
$$= \frac{\theta_{trn}^2\beta^2}{\tau_1^2}\mathrm{Tr}(h^Th) + 2\frac{\theta_{trn}^3\|t\|^2\beta}{\tau_1^2}\mathrm{Tr}(h^Tk^TA_{trn}^\dagger) + \frac{\theta_{trn}^4\|t\|^4}{\tau_1^2}\mathrm{Tr}((A_{trn}^\dagger)^Tkk^TA_{trn}^\dagger).$$

Where the last inequality is true due to the fact that $\|u\|^2 = 1$. How about when $N_{trn} < M$. Then we have the following string of equalities instead.

$$
\begin{aligned}
\|W\|_F^2 &= \operatorname{Tr}(W^T W) \\
&= \operatorname{Tr}\left( \left( \frac{\theta_{trn}\beta}{\tau_2}uh + \frac{\theta_{trn}^2\|h\|^2}{\tau_2}us^T \right)^T \left( \frac{\theta_{trn}\beta}{\tau_2}uh + \frac{\theta_{trn}^2\|h\|^2}{\tau_2}us^T \right) \right) \\
&= \frac{\theta_{trn}^2\beta^2}{\tau_2^2}\operatorname{Tr}(h^T u^T uh) + 2\frac{\theta_{trn}^3\|h\|^2\beta}{\tau_2^2}\operatorname{Tr}(h^T u^T us^T) + \frac{\theta_{trn}^4\|h\|^4}{\tau_1^2}\operatorname{Tr}(su^T us^T) \\
&= \frac{\theta_{trn}^2\beta^2}{\tau_2^2}\operatorname{Tr}(h^T h) + 2\frac{\theta_{trn}^3\|h\|^2\beta}{\tau_2^2}\operatorname{Tr}(h^T s^T) + \frac{\theta_{trn}^4\|h\|^4}{\tau_2^2}\operatorname{Tr}(ss^T).
\end{aligned}
$$

$\square$

### C.4   Step 4: Estimate using random matrix theory.

**Lemma 5.** *Suppose $A$ is an $p$ by $q$ matrix such that the entries of $A$ are independent and have mean 0, variance $1/q$, and bounded fourth moment. Let $W_p = AA^T$ and let $W_q = A^T A$. Let $C = p/q$. Suppose $\lambda_p, \lambda_q$ are a random eigenvalue of $W_p, W_q$. Then*

*1. If $p < q$, then $\mathbb{E}\left[\frac{1}{\lambda_p}\right] = \frac{1}{1-C} + o(1)$.*

*2. If $p < q$, then $\mathbb{E}\left[\frac{1}{\lambda_p^2}\right] = \frac{1}{(1-C)^3} + o(1)$.*

*3. If $p < q$, then $\mathbb{E}\left[\frac{1}{\lambda_p^3}\right] = \frac{1}{(1-C)^5} + o(1)$.*

*4. If $p < q$, then $\mathbb{E}\left[\frac{1}{\lambda_p^4}\right] = \frac{C^2 + \frac{22}{6}c + 1}{(1-C)^7} + o(1)$.*

*5. If $p > q$, then $\mathbb{E}\left[\frac{1}{\lambda_q}\right] = \frac{C^{-1}}{1-C^{-1}} + o(1)$.*

*6. If $p > q$, then $\mathbb{E}\left[\frac{1}{\lambda_q^2}\right] = \frac{C^{-2}}{(1-C^{-1})^3} + o(1)$.*

*7. If $p > q$, then $\mathbb{E}\left[\frac{1}{\lambda_q^3}\right] = \frac{C^{-3}(1+C^{-1})}{(1-C^{-1})^5} + o(1)$.*

*8. If $p > q$, then $\mathbb{E}\left[\frac{1}{\lambda_q^4}\right] = \frac{C^{-4}(C^{-2} + \frac{22}{6}C^{-1} + 1)}{(1-C^{-1})^7} + o(1)$.*

*Proof.* Suppose $A$ is an $p$ by $q$ matrix such that the entries of $A$ are independent and have mean 0, variance $1/q$, and bounded fourth moment. Then we know that $W_p = AA^T$ is an $p$ by $p$ Wishart matrix with $c = C$. If we send $p, q$ to infinity such that $p/q$ remains constant, then we have the eigenvalue distribution $F_p$ converges to the Marchenko Pastur distribution $F$ in probability.

From Rao & Edelman (2008), we know there exists a bi variate polynomial $L(m, z) = czm^2 - (1 - c - z)m + 1$ such that the zeros of $L(m, z)$ given by $L(m(z), z)$ are such that

$$
m(z) = \int \frac{1}{\lambda - z} dF(\lambda) = \mathbb{E}_\lambda\left[\frac{1}{\lambda - z}\right].
$$

For the Marchenko-Pastur distribution, we have that for $z = 0$, we get that $m(z) = 1/(1 - c)$. Thus, for $\lambda_p$ is an eigenvalue value of $W_p$, we have that

$$
\mathbb{E}\left[\frac{1}{\lambda_p}\right] = \frac{1}{1 - c} + o(1).
$$

For $\mathbb{E}_\lambda\left[\frac{1}{(\lambda - z)^2}\right]$ we need to calculate $m'(0)$. Using the implicit function theorem, we know that

$$
m'(z) = -1\left(\frac{\partial L}{\partial m}(m(z), z)\right)^{-1}\frac{\partial L}{\partial z}(m(z), z).
$$

Here we can see that $\partial L/\partial m = 2czm + c + z - 1$. Thus, at $(1/(1-c), 0)$, this is equal to $c - 1$. Also $\partial L/\partial z = cm^2 + m$. Again at $(1/(1-c), 0)$ this is equal to $\frac{c}{(1-c)^2} + \frac{1}{1-c} = \frac{1}{(1-c)^2}$. Thus, we have that

$$m'(0) = \frac{1}{(1-c)^3}.$$

Similarly, using the implicit function formulation, we can calculate $m''(0)$ and $m'''(0)$.

On the other hand if $q < p$, then $W_q := A^T A$ is not a Wishart matrix here, because it is scaled by the wrong constant. However, multiplying it by $1/C$ gives us the correct scaling. Thus, $A^T A/C$ is a Wishart matrix with $c = 1/C$ Thus, for $\lambda_q$ is an eigenvalue value of $W_q$, we have that

$$\mathbb{E}\left[\frac{1}{\lambda_q}\right] = \frac{C^{-1}}{1 - C^{-1}} + o(1).$$

We can obtain the rest in a similar manner from the previous results. $\qquad\square$

**Lemma 6.** *Suppose $A$ is an $p$ by $q$ matrix such that the entries of $A$ are independent and have mean 0, variance $1/q$, and bounded fourth moment. Let $C = p/q$ and let $x, y$ be unit vectors in $p$, then*

*1.* $\mathbb{E}[Tr(x^T(AA^T)^\dagger x)] = \begin{cases} \frac{1}{1-C} + o(1) & p < q \\ \frac{q}{p}\frac{C^{-1}}{1-C^{-1}} + o(1) & p > q \end{cases}$.

*2.* $\mathbb{E}[Tr(x^T(AA^T)^\dagger(AA^T)^\dagger x)] = \begin{cases} \frac{1}{(1-C)^3} + o(1) & p < q \\ \frac{q}{p}\frac{C^{-2}}{(1-C^{-1})^3} + o(1) & p > q \end{cases}$.

*3.* $\mathbb{E}[Tr(y^T(A^T A)^\dagger y)] = \begin{cases} \frac{p}{q}\frac{1}{1-C} + o(1) & p < q \\ \frac{C^{-1}}{1-C^{-1}} + o(1) & p > q \end{cases}$.

*4.* $\mathbb{E}[Tr(y^T(A^T A)^\dagger(A^T A)^\dagger y)] = \begin{cases} \frac{p}{q}\frac{1}{(1-C)^3} + o(1) & p < q \\ \frac{C^{-2}}{(1-C^{-1})^3} + o(1) & p > q \end{cases}$.

*Proof.* Let $A = U\Sigma V^T$ be the SVD. Then we have that $(AA^T)^\dagger = U(\Sigma^2)^\dagger U^T$. Then since $A$ is bi-unitary invariant, we have that $U$ is a uniformly random unitary matrix. Thus, $a = x^T U$ is a uniformly random unit vector. Note with probability 1, the rank of $A$ is full and that the non-zero eigenvalues of $A^T A$ and $AA^T$ are the same.

If $p < q$, then we have that

$$\mathbb{E}[\mathrm{Tr}(x^T(AA^T)^\dagger x)] = \sum_{i=1}^{p} a_i^2 \frac{1}{\sigma_i^2}.$$

Using Lemma 5, we have that $\mathbb{E}[1/\sigma_i^2] = 1/(1-C) + o(1)$. Thus, we have that

$$\mathbb{E}[\mathrm{Tr}(x^T(AA^T)^\dagger x)] = \sum_{i=1}^{p} \frac{1}{p}\frac{1}{1-C} + o(1).$$

On the other hand, if $p > q$, from Lemma 5, we have that $\mathbb{E}[1/\sigma_i^2] = C^{-1}/(1 - C^{-1}) + o(1)$. Thus,

$$\mathbb{E}[\mathrm{Tr}(x^T(AA^T)^\dagger x)] = \sum_{i=1}^{q} \frac{1}{p}\frac{C^{-1}}{1 - C^{-1}} + o(1).$$

Similarly, if we had we looking at $\mathrm{Tr}(x^T(AA^T)^\dagger(AA^T)^\dagger x)$, we would have a $1/\sigma_i^4$ term instead. Thus, if $p < q$, we would have that

$$\mathbb{E}[\mathrm{Tr}(x^T(AA^T)^\dagger(AA^T)^\dagger x)] = \frac{1}{(1-C)^3} + o(1).$$

A similar calculation holds for the others. $\qquad\square$

Now we have the following Lemma in the main text. However, here instead of having one big proof, we will separate each term out into its own lemma.

**Lemma 7.** *If $A_{trn}$ satisfies the standard noise assumptions, then we have that*

1. $\mathbb{E}[\beta] = 1 + o(1)$ *and* $\operatorname{Var}(\beta) = \frac{\theta_{trn}^2 c}{(max(M, N_{trn})|1-c|))} + o(1)$.

2. *If $c < 1$, then* $\mathbb{E}[\|h\|^2] = \frac{c^2}{1-c} + o(1)$ *and* $\operatorname{Var}(\|h\|^2) = \frac{c^3(2+c)}{N_{trn}(1-c)^3} + o(1)$.

3. *If $c > 1$, then* $\mathbb{E}[\|h\|^2] = \frac{c}{c-1} + o(1)$ *and* $\operatorname{Var}(\|h\|^2) = \frac{c^2(2c-1)}{N_{trn}(c-1)^3} + o(1)$.

4. $\mathbb{E}[\|k\|^2] = \frac{c}{1-c} + o(1)$ *and* $\operatorname{Var}(\|k\|^2) = \frac{c^2(2+c)}{M(1-c)^3} + o(1)$.

5. $\mathbb{E}[\|s\|^2] = \frac{c-1}{c} + o(1)$ *and* $\operatorname{Var}(\|s\|^2) = 2\frac{1}{Mc} + o(1)$

6. $\mathbb{E}[\|t\|^2] = 1 - c + o(1)$, $\operatorname{Var}(\|t\|^2) = 2\frac{c}{N_{trn}} + o(1)$.

**Lemma 11.** *$\beta$ term.*

*Proof.* First, we calculate the expected value of $\beta$. To do so, let $A_{trn} = U\Sigma V^T$ be the SVD. Then since $A_{trn}$ is bi-unitarily invariant, we have that $U, V$ are uniformly random unitary matrices. Since $u, v_{trn}$ are fixed. We have that $a := v_{trn}^T V \in \mathbb{R}^{N trn}$ and $b := U^T u \in \mathbb{R}^M$ are uniformly random unit vectors. In particular, we have that $\mathbb{E}[a_i] = 0, \mathbb{E}[b_i] = 0, \operatorname{Var}(a_i) = 1/N_{trn}, \operatorname{Var}(b_i) = 1/M$.

Thus, if $\sigma_i$ are the singular values for $A_{trn}$, then we have that

$$\beta = 1 + \theta_{trn} \sum_{i=1}^{\min(M, N_{trn})} \frac{1}{\sigma_i} a_i b_i.$$

Thus, if you take the expectation you get that

$$\mathbb{E}[\beta] = 1.$$

On the other hand, lets look at the variance. For the variance, we need to compute $\mathbb{E}[\beta^2]$. Now if we let $T := \theta_{trn} v_{trn}^T A_{trn}^\dagger u$. Then we have that

$$\beta^2 = 1 + T^2 + 2T.$$

Thus, again if we take the expectation, we get that

$$\mathbb{E}[\beta^2] = 1 + \mathbb{E}[T^2].$$

Again due to the fact that $a, b$ are independent have have mean 0 entries, the cross terms in $\mathbb{E}[T^2]$. Thus, we have that

$$\mathbb{E}[T^2] = \theta trn^2 \mathbb{E}\left[\sum_{i=1}^{\min(M, N_{trn})} \frac{1}{\sigma_i^2} a_i^2 b_i^2\right] = \theta trn^2 \frac{1}{MN_{trn}} \mathbb{E}\left[\sum_{i=1}^{\min(M, N_{trn})} \frac{1}{\sigma_i^2}\right].$$

Now we need to case on whether $M > N_{trn}$ or $M < N_{trn}$. Now to use Lemma 5, we note that $q = M$ and $p = N_{trn}$.

Suppose we have that $M > N_{trn}$, then in this case, we have that $q > p$. Thus, we have that

$$\mathbb{E}\left[\frac{1}{\sigma_i^2}\right] = \frac{1}{1-C} + o(1),$$

where $C = p/q = N_{trn}/M = 1/c$. Thus, we have that

$$\mathbb{E}\left[\frac{1}{\sigma_i^2}\right] = \frac{1}{1-1/c} + o(1) = \frac{c}{c-1} + o(1).$$

Thus, we have that

$$\mathbb{E}[T^2] = \theta_{trn}^2 \frac{c}{M(c-1)} + o\left(\frac{1}{M}\right).$$

Thus, we have

$$\mathrm{Var}(\beta) = \theta_{trn}^2 \frac{c}{M(c-1)} + o\left(\frac{1}{M}\right).$$

On the other hand, if $M < N_{trn}$. Then we have that $q < p$. Thus, we have that

$$\mathbb{E}\left[\frac{1}{\sigma_i^2}\right] = \frac{C^{-1}}{1 - C^{-1}} + o(1),$$

where $C = p/q = N_{trn}/M = 1/c$. Thus, we have that

$$\mathbb{E}\left[\frac{1}{\sigma_i^2}\right] = \frac{c}{1-c} + o(1).$$

Thus, we have that

$$\mathbb{E}[T^2] = \theta_{trn}^2 \frac{1}{N_{trn}} \left(\frac{c}{1-c} + o(1)\right) = \frac{c}{N_{trn}(1-c)} + o\left(\frac{1}{N_{trn}}\right).$$

Thus, we have

$$\mathrm{Var}(\beta) = \theta_{trn}^2 \frac{c}{N_{trn}(1-c)} + o\left(\frac{1}{N_{trn}}\right).$$

$\square$

**Lemma 12.** $\|h\|^2$ *term.*

*Proof.* We want to do a calculation similar to that in Lemma 1. Here we have that

$$\|h\|^2 = \mathrm{Tr}(h^T h) = \mathrm{Tr}((A_{trn}^\dagger)^T v_{trn} v_{trn}^T A_{trn}^\dagger) = \mathrm{Tr}(v_{trn}^T A_{trn}^\dagger (A_{trn}^\dagger)^T v_{trn}) = \mathrm{Tr}(v_{trn}^T (A_{trn}^T A_{trn})^\dagger v_{trn}).$$

To use Lemma 6, we note that $A = A_{trn}^T$, $q = M$, $p = N_{trn}$. Let us now suppose that $M < N_{trn}$. Then again taking the expectation, we see that

$$\mathbb{E}[\|h\|^2] = \frac{M}{N_{trn}}\left(\frac{c}{1-c} + o(1)\right) = \frac{c^2}{1-c} + o(1).$$

For the expectation of $\|h\|^4$, let $A_{trn} = U\Sigma V^T$ be the svd. Then $h = v_{trn}^T V\Sigma^\dagger U^T$. Let $a = v_{trn}^T V$ and note that $a$ is a uniformly random unit vector. Thus, we have that

$$\|h\|^2 = \sum_{i=1}^{M} \frac{1}{\sigma_i^2} a_i^2.$$

For the expectation of $\|h\|^4$, we note that

$$\|h\|^4 = \sum_{i=1}^{M} \sum_{j=1}^{M} \frac{1}{\sigma_i^2 \sigma_j^2} a_i^2 a_j^2 = \sum_{i=1}^{M} \frac{1}{\sigma_i^4} a_i^4 + \sum_{i \neq j} \frac{1}{\sigma_i^2} \frac{1}{\sigma_j^2} a_i^2 a_j^2.$$

Taking the expectation of the first term, we get

$$\sum_{i=1}^{M} \mathbb{E}\left[\frac{1}{\sigma_i^4}\right] \mathbb{E}[a_i^4] = \frac{3M}{N_{trn}(N_{trn}+2)} \left(\frac{c^2}{(1-c)^3} + o(1)\right) = 3\frac{c^3}{N_{trn}(1-c)^3} + o(1).$$

Taking the expectation of the second term, we get

$$M(M-1)\mathbb{E}\left[\frac{1}{\sigma_i^2}\right]^2 \mathbb{E}[a_i^2 a_j^2] = M(M-1)\frac{1}{N_{trn}(N_{trn}+2)}\left(\frac{c^2}{(1-c)^2}+o(1)\right) = \frac{c^4}{(1-c)^2} - \frac{c^3}{N_{trn}(1-c)^2}+o(1).$$

Thus, we have that

$$\mathbb{E}[\|h\|^4] = \frac{c^4}{(1-c)^2} + \frac{c^3(2+c)}{N_{trn}(1-c)^3} + o(1).$$

Thus, the variance is

$$\mathrm{Var}(\|h\|^2) = \frac{c^3(2+c)}{N_{trn}(1-c)^3} + o(1).$$

For $M > N_{trn}$, we instead have that

$$\mathbb{E}[\|h\|^2] = \frac{N_{trn}}{N_{trn}}\left(\frac{c}{c-1}+o(1)\right) = \frac{c}{c-1} + o(1).$$

For the expectation of $\|h\|^4$, we note that

$$\|h\|^4 = \sum_{i=1}^{N_{trn}}\sum_{j=1}^{N_{trn}}\frac{1}{\sigma_i^2\sigma_j^2}a_i^2 a_j^2 = \sum_{i=1}^{N_{trn}}\frac{1}{\sigma_i^4}a_i^4 + \sum_{i\neq j}\frac{1}{\sigma_i^2}\frac{1}{\sigma_j^2}a_i^2 a_j^2.$$

Taking the expectation of the first term, we get

$$\sum_{i=1}^{N_{trn}}\mathbb{E}\left[\frac{1}{\sigma_i^4}\right]\mathbb{E}[a_i^4] = \frac{3N_{trn}}{N_{trn}^2}\left(\frac{c^3}{(c-1)^3}+o(1)\right) = 3\frac{c^3}{N_{trn}(c-1)^3}+o(1).$$

Taking the expectation of the second term, we get

$$N_{trn}(N_{trn}-1)\mathbb{E}\left[\frac{1}{\sigma_i^2}\right]^2\mathbb{E}[a_i^2]^2 = N_{trn}(N_{trn}-1)\frac{1}{N_{trn}^2}\left(\frac{c^2}{(c-1)^2}+o(1)\right)$$

$$= \frac{c^2}{(c-1)^2} - \frac{c^2}{N_{trn}(c-1)^2}+o(1).$$

Thus, we have that

$$\mathbb{E}[\|h\|^4] = \frac{c^2}{(c-1)^2} + 3\frac{c^3}{N_{trn}(c-1)^3} - \frac{c^2}{N_{trn}(c-1)^2}+o(1) = \frac{c^2}{(c-1)^2} + \frac{c^2(2c-1)}{N_{trn}(c-1)^3}+o(1).$$

Thus, the variance is

$$\mathrm{Var}(\|h\|^2) = \frac{c^2(2c-1)}{N_{trn}(c-1)^3}+o(1).$$

$\square$

**Lemma 13.** $\|k\|^2$ *term.*

*Proof.* First note that $k$ only appears in the formula when $c < 1$. Thus, we can focus on this case. As with $h$, we have that

$$\|k\|^2 = \mathrm{Tr}(u^T(A_{trn}^\dagger)^T A_{trn}^\dagger u) = \mathrm{Tr}(u^T(A_{trn}A_{trn}^T)^\dagger u).$$

Again using Lemma 6, with $q = M, p = N_{trn}, A = A_{trn}, y = u$. Thus, since we have $q = M < N_{trn} = p$, we get that

$$\mathbb{E}[\|k\|^2] = \frac{c}{1-c}+o(1).$$

To calculate the variance, we need to calculate the expectation of $\|k\|^4$. Here be again let $A = U\Sigma V^T$ be the SVD. Then let $b := U^T u$. Then we have that

$$\|k\|^2 = \sum_{i=1}^{M} \frac{1}{\sigma_i^2} b_i^2.$$

Thus, we see that

$$\|k\|^4 = \sum_{i=1}^{M} \frac{1}{\sigma_i^4} b_i^4 + \sum_{i \neq j} \frac{1}{\sigma_i^2} \frac{1}{\sigma_j^2} b_i^2 b_j^2.$$

Taking the expectation of the first term we get

$$3 \frac{M}{M^2} \frac{c^2}{(1-c)^3} + o(1) = \frac{3c^2}{M(1-c)^3} + o(1).$$

Taking the expectation of the second term we get

$$\frac{M(M-1)}{M^2} \frac{c^2}{(1-c)^2} + o(1) = \frac{c^2}{(1-c)^2} - \frac{c^2}{M(1-c)^2} + o(1).$$

Thus, we have that

$$\mathbb{E}[\|k\|^4] = \frac{c^2}{(1-c)^2} + \frac{c^2(2+c)}{M(1-c)^3} + o(1).$$

Thus, we have that

$$\mathrm{Var}(\|k\|^2) = \frac{c^2(2+c)}{M(1-c)^3} + o(1).$$

$\square$

**Lemma 14.** $\|s\|^2$ *term.*

*Proof.* First, we note that $s$ only appears when $M > N_{trn}$. Thus, we only need to deal with that case. For this term, we note that $(I - A_{trn}A_{trn}^\dagger)$ is a projection matrix onto a uniformly random $M - N_{trn}$ dimensional subspace. Here be again let $A = U\Sigma V^T$ be the SVD. Then let $b := U^T u$.

$$\mathbb{E}[\|s\|^2] = \mathbb{E}[u^T u - u^T A_{trn} A_{trn}^\dagger u] = \mathbb{E}\left[1 - b^T \begin{bmatrix} I_{N_{trn}} & 0 \\ 0 & 0 \end{bmatrix} b\right] = 1 - \sum_{i=1}^{N_{trn}} \frac{1}{M} + o(1) = 1 - \frac{1}{c} + o(1)$$

Similarly, we have that

$$\|s\|^4 = \left(1 - \sum_{i=1}^{N_{trn}} b_i^2\right)^2$$

$$= 1 + \left(\sum_{i=1}^{N_{trn}} b_i^2\right)^2 - 2 \sum_{i=1}^{N_{trn}} b_i^2$$

$$= 1 + \sum_{i=1}^{N_{trn}} b_i^4 + \sum_{i \neq j}^{N_{trn}} b_i^2 b_j^2 - 2 \sum_{i=1}^{N_{trn}} b_i^2$$

Taking the expectation, we get that

$$\mathbb{E}[\|s\|^4] = 1 + 3\sum_{i=1}^{N_{trn}} \frac{1}{M^2} + \sum_{i \neq j}^{N_{trn}} \frac{1}{M^2} - 2\sum_{i=1}^{N_{trn}} \frac{1}{M} + o(1)$$

$$= 1 + \frac{3}{cM} + \frac{N_{trn}(N_{trn}-1)}{M^2} - 2\frac{1}{c} + o(1)$$

$$= 1 + \frac{3}{cM} + \frac{1}{c^2} - \frac{1}{cM} - 2\frac{1}{c} + o(1)$$

$$= \left(1 - \frac{1}{c}\right)^2 + \frac{2}{cM} + o(1)$$

Thus, we have that

$$\text{Var}(\|s\|^2) = 2\frac{1}{cM} + o(1)$$

$\square$

**Lemma 15.** $\|t\|^2$ *term.*

*Proof.* First, we note that $t$ only appears when $M < N_{trn}$. Thus, we only need to deal with that case. For this term, we note that $(I - A_{trn}^\dagger A_{trn})$ is a projection matrix onto a uniformly random $N_{trn} - M$ dimensional subspace. Then similar to $\|s\|^2$, we have that

$$\mathbb{E}[\|t\|^2] = \mathbb{E}[v_{trn}^T v_{trn} - v_{trn}^T A_{trn}^\dagger A_{trn} v_{trn}] = \mathbb{E}\left[1 - a^T \begin{bmatrix} I_M & 0 \\ 0 & 0 \end{bmatrix} a\right] = 1 - \sum_{i=1}^M \frac{1}{N_{trn}} + o(1) = 1 - c + o(1)$$

Similarly, we have that

$$\|t\|^4 = \left(1 - \sum_{i=1}^M a_i^2\right)^2$$

$$= 1 + \left(\sum_{i=1}^M a_i^2\right)^2 - 2\sum_{i=1}^M a_i^2$$

$$= 1 + \sum_{i=1}^M a_i^4 + \sum_{i \neq j}^M a_i^2 a_j^2 - 2\sum_{i=1}^M a_i^2$$

Taking the expectation, we get that

$$\mathbb{E}[\|t\|^4] = 1 + 3\sum_{i=1}^M \frac{1}{N_{trn}^2} + \sum_{i \neq j}^M \frac{1}{N_{trn}^2} - 2\sum_{i=1}^M \frac{1}{N_{trn}} + o(1)$$

$$= 1 + \frac{3c}{N_{trn}} + \frac{N_{trn}(N_{trn}-1)}{M^2} - 2c + o(1)$$

$$= 1 + \frac{3c}{N_{trn}} + c^2 - \frac{c}{N_{trn}} - 2c + o(1)$$

$$= (1-c)^2 + \frac{2}{cM} + o(1)$$

Thus, we have that

$$\text{Var}(\|t\|^2) = 2\frac{c}{N_{trn}} + o(1)$$

$\square$

Now we could just use the the fact that $|\mathbb{E}[XY] - \mathbb{E}[X]\mathbb{E}[Y]| < \sqrt{\mathrm{Var}(X)\mathrm{Var}(Y)}$. Another way to do this is via using big $O$ in probability. Which is defined as follows:

**Definition 3.** *We save that a sequence of random variables $X_n$ is $O_P(a_n)$, if there exists an $N$ such that for all $\epsilon > 0$, there exists a constant $L$ such that for all $n \geq N$, we have that $\mathrm{Pr}[|X_n| > La_n] < \epsilon$.*

Then the trace terms.

**Lemma 8.** *Under standard noise assumptions, we have that*

$$\mathbb{E}[\,Tr(h^T k^T A^\dagger_{trn})] = 0$$

*and*

$$\mathrm{Var}(\,Tr(h^T k^T A^\dagger_{trn})) = \chi_3(c)/N_{trn} + o(1),$$

*where $\chi_3(c) = \mathbb{E}[1/\lambda^3]$, $\lambda$ is an eigenvalue for $AA^T$ and $A$ is as in Lemma 6.*

*Proof.* First we note that

$$\mathrm{Tr}(h^T k^T A^\dagger_{trn}) = \mathrm{Tr}((A^\dagger_{trn})^T v_{trn} u^T (A^\dagger_{trn})^T A^\dagger_{trn}) = u^T (A^\dagger_{trn})^T (A^\dagger_{trn} A^\dagger_{trn})^T v_{trn}).$$

Again let $A_{trn} = U\Sigma V^T$ be the SVD. Then, we have the middle terms depending on $A_{trn}$ simplifies to

$$(A^\dagger_{trn})^T A^\dagger_{trn} (A^\dagger_{trn})^T = U(\Sigma^\dagger)^T \Sigma^\dagger (\Sigma^\dagger)^T V^T.$$

Thus, again letting $b = u^T U$ and $a = V^T v_{trn}$. We see that

$$\mathrm{Tr}(h^T k^T A^\dagger_{trn}) = \sum_{i=1}^{M} a_i b_i \frac{1}{\sigma_i^3}.$$

Now if take the expectation, since $a, b$ are independent and mean 0, we see that

$$\mathbb{E}_{A_{trn}}[\mathrm{Tr}(h^T k^T A^\dagger_{trn})] = 0.$$

Let us also compute the variance. Here we have that

$$\mathbb{E}[\mathrm{Tr}(h^T k^T A^\dagger_{trn})^2] = \sum_{i=1}^{M} \mathbb{E}\left[\frac{1}{\sigma_i^6}\right] \mathbb{E}[a_i^2]\mathbb{E}[b_i^2] + 0.$$

Now for the Marchenko Pastur distribution we have that the expectation of $1/\lambda^3 = \chi_3(c)$. where $\chi_3$ is some function. Thus, we have that

$$\mathbb{E}[\mathrm{Tr}(h^T k^T A^\dagger_{trn})^2] = \frac{1}{N_{trn}}\chi_3(c) + o(1).$$

$\square$

**Lemma 9.** *Under standard noise assumptions, we have that*

$$Tr((A^\dagger_{trn})^T k k^T A^\dagger_{trn}) = \frac{c^2}{(1-c)^3} + o(1)$$

*and*

$$\mathrm{Var}(\,Tr((A^\dagger_{trn})^T k k^T A^\dagger_{trn})) = \frac{3}{M}\chi_4(c) - \frac{1}{M}\frac{c^4}{(1-c)^6} + o(1)$$

*where $\chi_4(c) = \mathbb{E}[1/\lambda^4]$, $\lambda$ is an eigenvalue for $AA^T$ and $A$ is as in Lemma 6.*

*Proof.* Now using Lemma 6, we see that

$$\mathbb{E}_{A_{trn}}[\text{Tr}((A_{trn}^{\dagger})^T k k^T A_{trn}^{\dagger})] = \frac{c^2}{(1-c)^3} + o(1).$$

Similar to proofs before, we have that

$$\mathbb{E}_{A_{trn}}[\text{Tr}((A_{trn}^{\dagger})^T k k^T A_{trn}^{\dagger})^2] = \sum_{i=1}^{M} \frac{3}{M^2} \chi_4(c) + \sum_{i \neq j} \frac{1}{M^2} \frac{c^4}{(1-c)^6} + o(1).$$

Where $\chi_4(c) = \mathbb{E}[1/\lambda^4]$ for the Marchenko Pastur distribution. Thus, we have that

$$\text{Var}(\text{Tr}((A_{trn}^{\dagger})^T k k^T A_{trn}^{\dagger})) = \frac{3}{M} \chi_4(c) + \frac{1}{M} \frac{c^4}{(1-c)^6} + o(1).$$

$\square$

**Lemma 10.** *Under the same assumptions as Proposition 2, we have that $\text{Tr}(h^T s^T) = 0$.*

*Proof.* Here we note that $h^T = (A_{trn}^{\dagger})^T v_{trn}$ and $s^T = u^T (I - A_{trn} A_{trn}^{\dagger})^T$. Thus, we have that

$$\begin{aligned}
\text{Tr}(h^T s^T) &= \text{Tr}((A_{trn}^{\dagger})^T v_{trn} u^T - (A_{trn}^{\dagger})^T v_{trn} u^T (A_{trn} A_{trn}^{\dagger})^T) \\
&= \text{Tr}(v_{trn}^T A_{trn}^{\dagger} u) - \text{Tr}(u^T (A_{trn} A_{trn}^{\dagger})^T (A_{trn}^{\dagger})^T v_{trn}) \\
&= \text{Tr}(v_{trn}^T A_{trn}^{\dagger} u) - \text{Tr}(v_{trn}^T A_{trn}^{\dagger} A_{trn} A_{trn}^{\dagger} u) \\
&= \text{Tr}(v_{trn}^T A_{trn}^{\dagger} u) - \text{Tr}(v_{trn}^T A_{trn}^{\dagger} u) \\
&= 0
\end{aligned}$$

$\square$

As we can see that if we take the expectation of $\|W\|$ over $A_{trn}$, since the variance of each of the terms is small, we can approximate $\mathbb{E}[XY]$ with $\mathbb{E}[X]\mathbb{E}[Y]$. Then we get the following.

If $M < N_{trn}$, we have that

$$\begin{aligned}
\mathbb{E}_{A_{trn}}[\|W\|^2] &= \frac{\theta_{trn}^2}{(1+\theta_{trn}^2 c)^2} \frac{c^2}{(1-c)} + 0 + \frac{\theta_{trn}^4 (1-c)^2}{(1+\theta_{trn}^2 c)^2} \frac{c^2}{(1-c)^3} \\
&= c^2 \frac{\theta_{trn}^2 + \theta_{trn}^4}{(1+\theta_{trn}^2 c)^2 (1-c)}.
\end{aligned}$$

On the other hand, $M > N_{trn}$, we have that

$$\begin{aligned}
\mathbb{E}_{A_{trn}}[\|W\|^2] &= \frac{\theta_{trn}^2}{(1+\theta_{trn}^2)^2} \frac{c}{c-1} + \frac{\theta_{trn}^4}{(1+\theta_{trn}^2)^2} \frac{c^2}{(c-1)^2} \frac{c-1}{c} \\
&= \frac{c}{c-1} \frac{\theta_{trn}^2 (1+\theta_{trn}^2)}{(1+\theta_{trn}^2)^2} \\
&= \frac{\theta_{trn}^2}{1+\theta_{trn}^2} \frac{c}{c-1}.
\end{aligned}$$

Now combining everything together, we get that

$$\mathbb{E}_{A_{trn},A_{tst}} \left[ \frac{\|\theta_{tst} X_{tst} - W(\theta_{tst} X_{tst} + A_{tst})\|}{N_{tst}} \right] = \begin{cases} \frac{\theta_{tst}^2}{N_{tst}(1+\theta_{trn}^2 c)^2} + \frac{1}{M} c^2 \frac{\theta_{trn}^2 + \theta_{trn}^4}{(1+\theta_{trn}^2 c)^2 (1-c)} & c < 1 \\ \frac{\theta_{tst}^2}{N_{tst}(1+\theta_{trn}^2 c)^2} + \frac{1}{M} \frac{\theta_{trn}^2}{1+\theta_{trn}^2} \frac{c}{c-1} & c > 1 \end{cases}.$$

### C.5 Proof of Theorem

We can see that the main text has how to put all of the pieces together to prove the main Theorem. We don't replicate that here.

### C.6 Formula for $\hat{\theta}_{opt-trn}$

As stated in the main text, we only need to take the derivative. So, we don't present that calculation here as it is fairly straightforward.

## D Generalizations

In this section we discuss some possible generalizations of the method.

### D.1 Higher rank

Let us present some heuristics for the higher rank formula. To do so we shall need some notation. Let $X_{trn} = \sum_{i=1}^{r} \sigma_i^{trn} u_i (v_i^{trn})^T$. Let $A$ be the noise matrix. Then for $1 \leq j \leq r$, define

$$A_j = \left( A + \sum_{i=1}^{j-1} \sigma_i^{trn} u_i (v_i^{trn})^T \right)$$

We shall now make some assumptions. Specifically, we assume that $u_j, v_j^{trn}$, and $A_j$ are all such that for $i_1 \neq i_2$, and for all $j$ we have that

$$\mathbb{E}[u_{i_1}^T A_j A_j^\dagger u_{i_2}] = \mathbb{E}[(v_{i_1}^{trn})^T A_j^\dagger A_j v_{i_2}^{trn}] = 0.$$

Additionally, we assume that for all $i_1, i_2, j$ we have that $\mathbb{E}[(v_{i_1}^{trn})^T A_j^\dagger u_{i_2}] = 0$. We also assume that the variance of these terms goes to 0 as $N_{trn}, M$ go to infinity.

**Lemma 16.** *With the given assumptions, we have that for all $i < j$,*

$$\sigma_i^{trn} u_i (v_i^{trn})^T A_j^\dagger \approx \sigma_i^{trn} u_i (v_i^{trn})^T A_{j-1}^\dagger \approx \sigma_i^{trn} u_i (v_i^{trn})^T A_{j-2}^\dagger \approx \ldots \approx \sigma_i^{trn} u_i (v_i^{trn})^T A_{i+1}^\dagger$$

*Proof.* Write $A_j = A_{j-1} + \sigma_j^{trn} u_j (v_k^{trn})^T$ and the use Meyer (1973) to expand the pseudoinverse of $A_j$. When we do this, we see that due to the assumption all terms expect $\sigma_i^{trn} u_i (v_i^{trn})^T A_{j-1}^\dagger$ are small. □

Define $h_j = (v_j^{trn})^T A_j^\dagger$, $k_j = \sigma_j^{trn} A_j^\dagger u_j$, $t_j = (v_j^{trn})^T (I - A_j^\dagger A_j)$, $s_j = \sigma_j^{trn} (I - A_j A_j^\dagger) u_j$, $\beta_j = 1 + \sigma_j^{trn} (v_j^{trn})^T A_j^\dagger u_j$, $\tau_1^{(j)} = \|t_j\|^2 \|k_j\|^2 + \beta_j^2$, $\tau_2^{(j)} = \|s_j\|^2 \|h_j\|^2 + \beta_j^2$, and similarly $p_1^{(j)}, p_2^{(j)}, q_1^{(j)}$, and $q_2^{(j)}$. Now, we can write

$$X_{trn} + A = \sigma_r^{trn} u_r (v_r^{trn})^T + A_{r-1}$$

Then we have that

$$W = X(\sigma_r^{trn} u_r (v_r^{trn})^T + A_r)^\dagger = \sum_{i=1}^{r} \sigma_i^{trn} u_i (v_i^{trn})^T (\sigma_r^{trn} u_r (v_r^{trn})^T + A_r)^\dagger$$

Expanding and using the lemma, we get that

$$W \approx \sum_{i=1}^{r} \sigma_i^{trn} u_i (v_i^{trn})^T A_{i+1}^\dagger = \begin{cases} \sum_{i=1}^{r} \frac{\sigma_i^{trn} \beta_i}{\tau_1^{(i)}} u_i h_i + \frac{(\sigma_i^{trn})^2 \|t_i\|^2}{\tau_1^{(i)}} u_i k_i^T A_i^\dagger & c < 1 \\ \sum_{i=1}^{r} \frac{\sigma_i^{trn} \beta_i}{\tau_2^{(i)}} u_i h_i + \frac{(\sigma_i^{trn})^2 \|h_i\|^2}{\tau_2^{(i)}} u_i s_i^T & c > 1 \end{cases}$$

Where the second equality comes from the rank 1 results.

Now that we have an approximation for $W$ (given our assumptions), we can now approximate the variance and bias terms again. Let $W_i$ denote the $i$th factor (corresponding to $u_i$) of $W$. First, for the bias, due to the orthogonality of the $u$'s we get that

$$\|X_{tst} - WX_{tst}\|_F^2 = \sum_{i=1}^r \left\| \sigma_i^{tst} u_i (v_i^{tst})^T - W_i \sum_{j=1}^r \sigma_i^{tst} u_i (v_i^{tst})^T \right\|_F^2$$

Again, using our assumptions, we see that the terms in the $j$ summation dropout besides when $j = i$. Then again using our rank 1 result, we get that

$$\|X_{tst} - WX_{tst}\|_F^2 = \sum_{i=1}^r \left( \frac{\beta_i}{\tau_{idx}^{(i)}} \sigma_i^{tst} \right)^2$$

For the variance, we again estimate the norm of $W$ by expanding the trace. Here we see that the cross terms are 0 due to factors of $u_{i_1}^T u_{i_2}$. For the diagonal terms, we again use the rank 1 results and get that

$$\|W\|_F^2 = \sum_{i=1}^r \frac{(\sigma_i^{trn})^2 \beta_i^2}{(\tau_1^{(i)})^2} \operatorname{Tr}(h_i^T h_i) + 2 \frac{(\sigma_i^{trn})^3 \|t_i\|^2 \beta_i}{(\tau_1^{(i)})^2} \operatorname{Tr}(h_i^T k_i^T A_i^\dagger) + \frac{(\sigma_i^{trn})^4 \|t_i\|^4}{(\tau_1^{(i)})^2} \operatorname{Tr}((A_i^\dagger)^T k_i k_i^T A_i^\dagger)$$

and if $c > 1$, then we have that

$$\|W\|_F^2 = \sum_{i=1}^r \frac{(\sigma_i^{trn})^2 \beta_i^2}{(\tau_2^{(i)})^2} \operatorname{Tr}(h_i^T h_i) + 2 \frac{(\sigma_i^{trn})^3 \|h_i\|^2 \beta_i}{(\tau_2^{(i)})^2} \operatorname{Tr}(h_i^T s_i^T) + \frac{(\sigma_i^{trn})^4 \|h_i\|^4}{(\tau_2^{(i)})^2} \operatorname{Tr}(s_i s_i^T).$$

The final step would be to estimate each of these terms using random matrix theory. However, unfortunately the $A_j$ may not satisfy all of the needed conditions. However, we know that $A_j$ is a perturbation of $A$ and $A$ satisfies all of the needed conditions. Hence, if the perturbation is small, we can replace $A_j$ with $A$ and hopefully not incur too much cost. Note this is also the reason why the previous assumptions might be reasonable. If we replace $A_j$'s with $A$ use our estimates from the rank 1 result. We then get our estimate for the generalization error for general rank $r$ data.

$$R(\theta_{trn}, \theta_{tst}, c, \Sigma_{trn}, \Sigma_{tst}) = \sum_{i=1}^r \frac{(\theta_{tst}\sigma_i^{tst})^2}{N_{tst}(1 + (\theta_{trn}\sigma_i^{trn})^2 c)^2} + \frac{c^2((\theta_{trn}\sigma_i^{trn})^2 + (\theta_{trn}\sigma_i^{trn})^4)}{M(1 + (\theta_{trn}\sigma_i^{trn})^2 c)^2(1-c)} + o(1) \qquad (14)$$

and if $c > 1$, we have that

$$R(\theta_{trn}, \theta_{tst}, c, \Sigma_{trn}, \Sigma_{tst}) = \sum_{i=1}^r \frac{(\theta_{tst}\sigma_i^{tst})^2}{N_{tst}(1 + (\theta_{trn}\sigma_i^{trn})^2)^2} + \frac{c(\theta_{trn}\sigma_i^{trn})^2}{M(1 + (\theta_{trn}\sigma_i^{trn})^2)(c-1)} + o(1). \qquad (15)$$

In the experimental section, we see that for small values of $r$ for $c$ bounded away from 1. This seems to be good estimate for the generalization error.

## E Experiments

Please see accompanying notebook for code to produce the data for all of the figures.

### E.1 Low SNR and High SNR data

For low SNR data, we sample the $\theta$ times singular values from a squared standard Gaussian. We do this independently for all $2r$ singular values. We call this the low SNR region because $\theta$ is not being scaled with the number of data points. Hence as $N_{trn}, N_{tst} \to \infty$, the SNR goes to 0.

For the high rank data, we sample $\theta$ times singular values from a squared Gaussian and then multiply by $\sqrt{N_{trn}}, \sqrt{N_{tst}}$. Hence here the SNR does not go to 0 as $N_{trn}, N_{tst} \to \infty$.

## F   Generalization Error versus Training noise level plots

### F.1   More Tests for Rank 1

Here we provide more examples of $c$ and how our theoretical formula matches the experimental performance exactly.

Each empirical point is the average over 50 trials. These were run on a laptop with 8gb of RAM and an i3 processors. The average time to produce any of these plots is about 10 to 30 minutes.

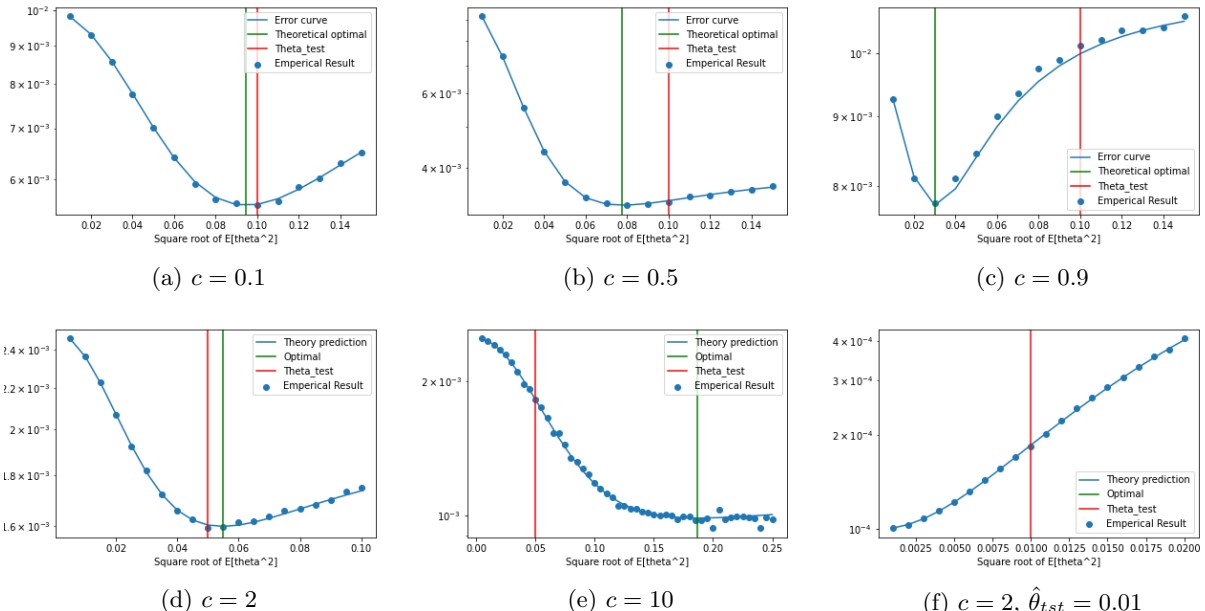

Figure 8: Figures (a) - (e) showing the accuracy of the formula for the expected mean squared error for $c = 0.1, 0.5, 0.9, 2, 10$ for fixed value of $\hat{\theta}_{tst}$. Figure (f) empirically verifies the existence of a regime where training on pure noise is optimal. Here the red and green lines represent $\mathbb{E}[\hat{\theta}_{tst}^2]$ and $\mathbb{E}[\hat{\theta}_{trn}^2]$ respectively. Each empirical data point is averaged over at least 50 trials.

### F.2   Rank 2 Data

Let us now demonstrate that the double descent shaped curve exists beyond rank 1 data and linear autoencoders. We will do this by gradually making the set up more complicated until we can no longer recreate this phenomena. First, we consider rank 2 data is of the following form. Let $W_{data}$ be some fixed matrix, then our data is generated by

$$X = \texttt{relu}(W_{data}\texttt{relu}(uv^T).$$

Where a different $v$ is sampled for the training and test data. the results for this can be seen in Figure 9. As we can from the figure, we have the exact same qualitative trend for $c$ that we saw before. That is, as $c$ goes from 0 to 1, we have that $\hat{\theta}_{trn}$ goes from $\hat{\theta}_{tst}$ to 0, and then as $c \to \infty$, we have that $\hat{\theta}_{trn}$ goes to infinity as well.

### F.3   MNIST Data

We now look at the linear network with MNIST data.

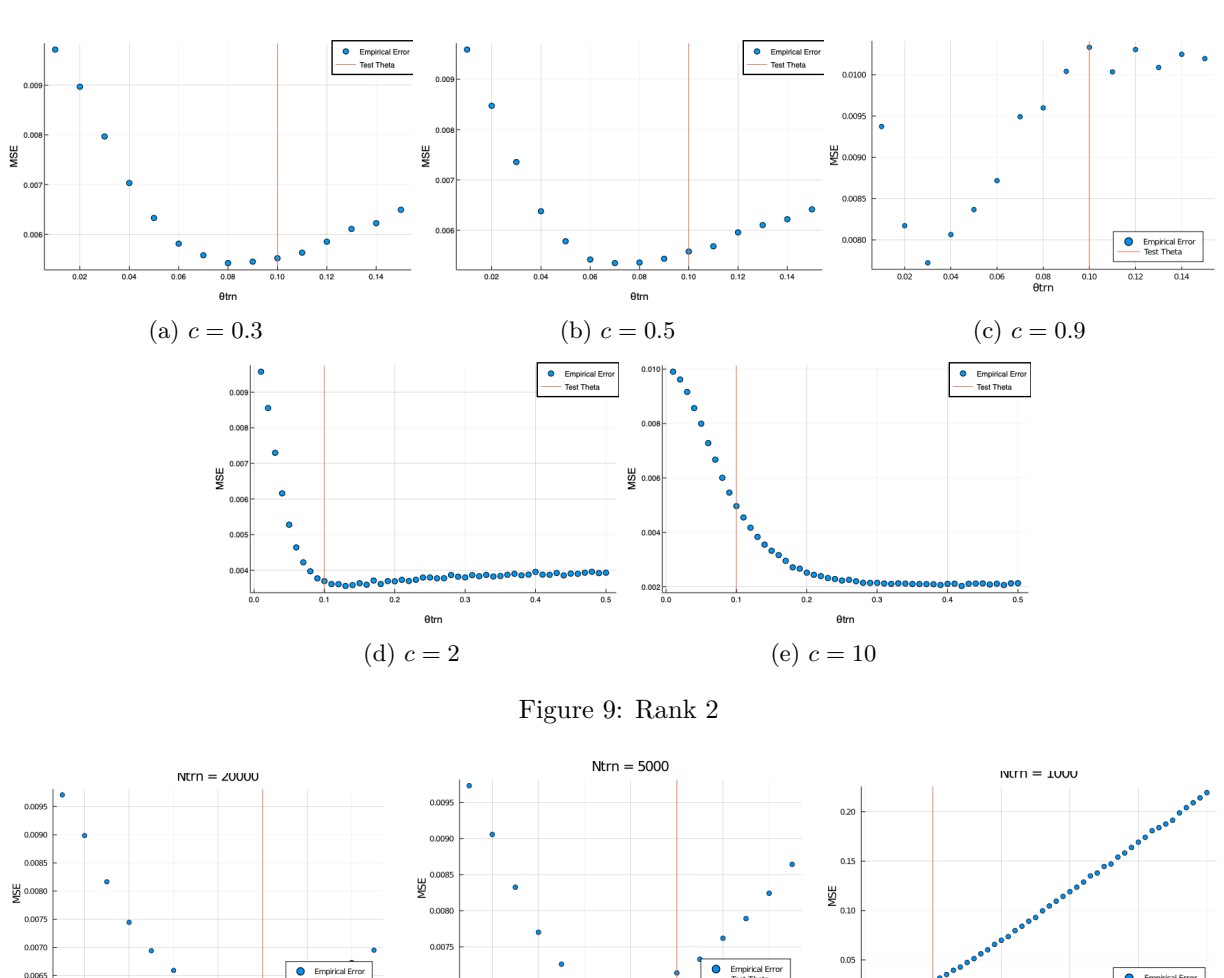

Figure 9: Rank 2

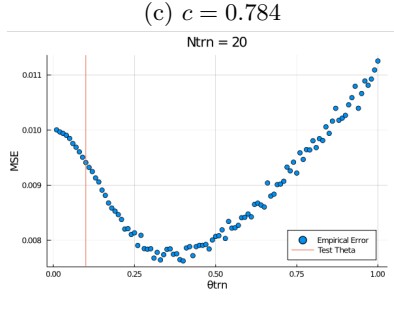

Figure 10: MNIST

### F.3.1 Non-linear Network

Here, we trained each network for 1500 epochs. During each epoch we computed a gradient using the whole data set. We used Adam as the optimizer with the code written in Pytorch. Each data point was generated over 20 trials. These experiments take a little bit more time to run and the one with bigger amounts of data can take upto 5 hours on a google cloud instance with 16gb RAM. Here we used a Telse P4 gpu. LRL is a model with a reLU at the end of the first layer only.

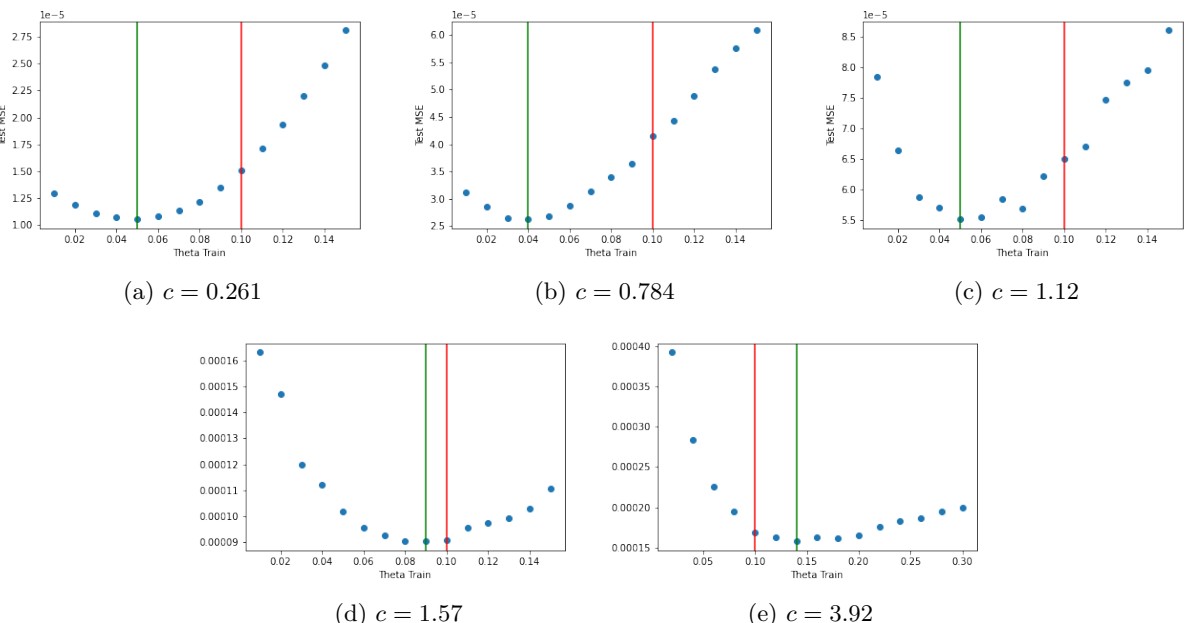

Figure 11: MNIST - LRL model

