# OpenReview forum: "Training Data Size Induced Double Descent For Denoising Feedforward Neural Networks and the Role of Training Noise"
_TMLR — Accepted by TMLR_

### Review · Reviewer_PutB · 2023-02-04

**Summary Of Contributions:**

The paper focuses on the double descent both empirically and theoretically of the denoising model. The main theorem provides an asymptotic result for the excess risk of the rank-one data, which borrows the techniques from random matrix theory.


**Audience:**

Yes

**Claims And Evidence:**

Yes

**Requested Changes:**

- In Figure 2, why does the phase transition not occur at 1?

- I suggest the authors set the assumptions more formally and mathematically, and put the discussion in the remark.

- The low-dimensional data assumption is ok for me. But I’m wondering that it  would be possible to consider the case that the target function is low-dimensional? This setting is more realistic.

- The empirical validation on the approximation in Eqs. (7) and (8) is needed.

- In theorem 1, how does the test error vary with N_{tst}, M? This requires more discussion. The comparison with previous work in double descent, e.g., least squares, random features, is needed. For example, when I saw the result in Theorem 1, it is similar to Hastie’s result. What is the same point and separation under these problem settings?


**Strengths And Weaknesses:**

Strength

- This paper gives a theoretical analyses under the rank-1 data and indicates how the optimal SNR affects the double descent curve.

Weakness

- The description on assumption is unclear.

- In theorem 1, the assumption $\theta_{trn} = O(\sqrt{N_{trn}})$ does not make sense as the dimension of $\theta_{trn}$ is the input dimension $d$.

---

> ### Author Response · Authors · 2023-02-12
> **Response to Comments**
>
> We thank the reviewer for their comments.
>
> 1. We have clarified in section 2 that $\theta_{trn}, \theta_{tst}$ are scalars and not vectors. Further, the $y$'s are vectors and not scalars.
>
> 2. In Figure 2, for the top row (for the linear model), it does occur at 1. So we assume the reviewer is asking for the bottom row. For the bottom row, we have used a three-layer neural network with non-linearities. For such a model, we do not know the exact interpolation threshold. Hence it need not be at one.
>
> 3. For the mathematical nature of the assumptions, we would like to ask a clarifying question. Which set of assumptions (sections 4.1, 4.2, 4.3 were not mathematical)
>
> 4. We agree that transitioning from the one-dimensional to the low dimensional case would make the work more interesting. However, theoretically analyzing this situation is challenging. We can see from the paper that even the rank 1 case is challenging.
>
> 5. The empirical validation for equations 7 and 8 can be seen in Figure 5 on page 8.
>
> 6. If we fix $\hat{\theta}\_{tst}$ (the test error SNR), the error is independent of $N\_{tst}$.
>
> Scaling $M$ is more subtle, as the quantity $c = M/N\_{trn}$ depends on $M$. We can see that if we increase $M$ while keeping $N\_{trn}$ fixed, this results in $c \to \infty$, and the variance term goes to 0, and we are left with just the bias term for the $c > 1$.
>
> However, if we scale $M$ and $N_{trn}$ while keeping $c$ fixed, then the variance term goes to zero because of the $1/M$ factor, and we have the bias term again.
>
> In both cases, the formula provides the leading of the asymptotics.

---

> > ### Comment · Reviewer_PutB · 2023-02-22
> > **concerns on assumptions and comparisons with Hastie's result**
> >
> > Thanks for your reply.
> >
> > I still care about the following issues:
> >
> > 1. The assumptions in sections 4.1, 4.2, 4.3 are not writing in a mathmatical way. I suggest the authors to follow the double descent theory references to organize the assumptions. In theorem 1, there is no clear assumptions.  For example,
> >
> > "We assume that each entry of the noise matrix A has mean 0, variance 1/M and that the entries of A are pairwise
> > uncorrelated. Additionally, we shall assume that A is rotationally bi-invariant." -> presented by equations and put it as an assumption.
> >
> > Besides, most of these description belong to problem settings instead of assumptions.
> >
> > 2. the discussion with Hastie's result is missing.

---

### Review · Reviewer_LAyJ · 2023-02-12

**Summary Of Contributions:**

This work considers the generalization error of a linear denoising model. For rank-1 data, it shows that the generalization error, as a function of the number of training samples, follows a double descent curve. In more general settings, experiments are conducted that also suggest a similar double descent curve. Moreover, the paper shows that the double descent phenomenon can be mitigated by optimally tuning the SNR in the denoising procedure. In other words, the generalization error of a denoising procedure with the optimally tuned SNR is a decreasing function of the number of training samples.

**Audience:**

Yes

**Claims And Evidence:**

Yes

**Requested Changes:**

First please polish the writing. Moreover, please consider commenting on my questions above to help me better understand the results.

**Strengths And Weaknesses:**

This work studies a pretty interesting topic: the generalization of a denoising model. However, even before finishing my first pass, I noticed the following clear weaknesses in the paper:

1. The writing could be improved. Too many typos. Also inconsistent tenses. The current writing quality really discourages me to spend more time reviewing the manuscript.

While the topic sounds to be interesting, and also novel to my knowledge, I am not sure I fully understand the problem setting. The mathematics could also be made more clear.

2. Could have emphasized that $\\theta$ is just a scalar instead of a matrix.
3. On page 3 bottom, what is the "expected norm of the noise distribution"?
4. In Section 4.1. If $U$, $V\_{tst}$, and $\\Sigma\_{tst}$ are all given and fixed, why should we care about denoising at all, as we already know the test data?
5. Remark 1 on the bottom of page 6: could you please be more specific? What is degenerate and why degeneration causes a problem?
6. Eq. (4), the expectation is taken over what randomness?
7. Overall, I do not see why the current denoising problem formulation makes sense. Firstly I would encourage the authors to explain what is known and what is unknown. If the test data is already given why would people be interested in denoising it? Secondly, I would encourage the authors to provide some *specific* examples that connect the mathematical formulation (e.g., eqs (1) and (2)) to practical denoising procedures.

I have more questions regarding the theoretical results.

8. In equations (5) and (6), the $o(1/N\_{tst})$ and $o(1/M)$ terms look suspicious. Could you please specify the hidden constants/factors?
9. The condition that $\\theta\_{tst}$ is $O(\\sqrt{N\_{tst}})$ is also hard to interpret. First, please specify the hidden constants/factors. Second, it seems to suggest that $\\theta\_{tst}$ could be a function of $N\_{tst}$, then if $\\theta\_{tst} = 1/{N\_{tst}}$ for example, it is not clear which term in eq (5) is the leading term.
10. When $c\\to 1$ the current bounds in eqs (5) and (6) are infinite. But this does not seem to be sharp. Note that under the assumption that $r=1$, the whole problem is effectively only a one-dimensional problem as $u$ is given. Then computing the eigenvalue of a (fixed) 1-dim projection of the noise matrix effectively reduces to computing the norm of a random vector. This is only a random variable and should be well-bounded (away from zero) no matter $c=1$ or not. So I do not think the test error should be unbounded when $c=1$.
11. On page 11 top, could you be more specific about the technical difficulty? Note that the Sherman-Morrison-Woodbury formula gives you the formula of the form $(A+XT)\^{\\dagger}$ for general matrix $A,X,Y$. I am not sure if it is necessary at all to restrict oneself to rank 1.

Given the above weaknesses, I am leaning toward rejecting the current version of the manuscript.

---

> ### Author Response · Authors · 2023-02-12
> **Weaknesses**
>
> We thank the reviewer for their feedback. Here we address their concerns and hope that with the changes, the reviewer votes for acceptance rather than rejection.
>
> 1. **Writing**
>
> We have updated the manuscript to a version in which the writing has been improved.
>
> 2. **$\theta$ is a scalar.**
>
> We explicitly clarify this in the setup in Section 2.
>
> 3. **Expected norm**
>
> This should be if $\xi \sim \mathcal{D}\_{noise}$ then $\mu\_{noise} = \mathbb{E}\_{\mathcal{D}\_{noise}}[\|\|\xi\|\|]$. We have clarified this in the updated manuscript.
>
> 4. **Denoising Problem**
>
> We thank the reviewer for pointing this out. We have fixed this in the updated manuscript. We only assume that $U, V_{trn}, V_{tst}$ are fixed and not that they are given. To clarify, the problem has access to $X_{trn}$ (noiseless training data), $Y_{tst}$ (noisy test data), and $\theta_{tst}$, SNR for the noisy test data.
>
> 5. **Degenerate**
>
> Here we specifically mean that if $X_{trn}$ is Gaussian and $A_{trn}$ is Gaussian, then $Y_{trn} = \theta_{trn}X_{trn} + A_{trn}$ is Gaussian. Hence the data does not have any structure different from the noise.
>
> 6. **Randomness**
>
> The randomness is over the training noise $A_{trn}$ and the test noise $A_{tst}$. We clarify this in the updated manuscript.
>
> 7. **Problem Setup**
>
> Hopefully, with the clarification to point (4), the problem setup makes more sense. We have noisy test data and noiseless training data and want to train a denoiser to denoise the test data. We are then interested in determining the optimal amount of noise that should be added to the training data when training the denoiser.

---

> ### Author Response · Authors · 2023-02-12
> **Theoretical Questions**
>
> We thank the reviewer for their questions. We hope that with the clarifications, the reviewer's view of our work is improved.
>
> 8. **Little o terms**
>
> We thank the reviewer for their comment, and we note that the $o(1/N_{tst})$ has been updated to $o(\theta_{tst}^2/N_{tst})$ in the updated manuscript.
>
> The little o terms come from the 4th step of the proof (shown in Figure 7, page 10). Specifically, the following displayed equation after Lemma 3 on page 11
>
> $$\mathbb{E}\_{A\_{tst}}\left[\frac{\|\theta\_{tst}X\_{tst}-WY\_{tst}\|_F^2}{N\_{tst}}\right] = \frac{1}{N\_{tst}}\frac{\beta^2}{\tau\_i^2}\theta\_{tst}^2 + \frac{1}{M}\|\|W\|\|\_F^2$$
>
> and the equations in Lemma 4 are **exact**. To further emphasize, these are not bounded or approximations but exact expressions. The error terms come **purely** from the 4th step of the proof.
>
> To do the estimates, we assume that the eigenvalues come from the Marchenko-Pastur distribution. The convergence rate to the Marchenko-Pastur distribution is discussed in the paragraph at the top of page 12.
>
> Then Lemma 5 on page 12 is the first term where the first estimate appears with the $o(1)$ term. This term propagates through so that we get
>
> $$\mathbb{E}\_{A\_{trn}}[\|W\|^2\_F] = \frac{c^2((\theta\_{trn}\sigma\_1^{trn})^2 + (\theta\_{trn}\sigma\_1^{trn})^4)}{(1+(\theta_{trn}\sigma\_1^{trn})^2c)^2(1-c)} + o(1) $$
>
> and
>
> $$\mathbb{E}\_{A_{trn}}\left[\frac{\beta^2}{\tau_1^2}\right] = \frac{(\sigma_1^{tst})^2}{(1+(\theta_{trn}\sigma_1^{trn})^2c)^2} + o(1)$$
>
> Then using
>
> $$\mathbb{E}\_{A\_{tst}}\left[\frac{\|\theta\_{tst}X\_{tst}-WY\_{tst}\|_F^2}{N\_{tst}}\right] = \frac{1}{N\_{tst}}\frac{\beta^2}{\tau\_i^2}\theta\_{tst}^2 + \frac{1}{M}\|\|W\|\|\_F^2$$
>
> To get the error, the little o terms.
>
> 9. **$\theta_{tst}$ scaling.**
>
> We recall that $Y_{tst} = \theta_{tst} X_{tst} + A_{tst}$, and these are $M \times N_{tst}$ matrices and the entries of $A_{tst}$ are drawn I.I.D.  from $\mathcal{N}(0,1/M)$. Further, from the assumptions in Section 4.1, we have that $\|\|X\_{tst}\|\|_F = 1$. However, we can see that $\mathbb{E}[\|\|A\_{tst}\|\|\_F] = \sqrt{N\_{tst})$.
>
> Thus, we allow $\theta\_{tst}$ to be $O(\sqrt{N\_{tst}})$. The constant here is $\theta\_{tst}/\sqrt{N\_{tst}}$, which, as mentioned in section 4.4, will be $\hat{\theta}\_{tst}$ which is the Signal to Noise Ratio.
>
> We though agree with the reviewer that the most interpretable case is when $\hat{\theta}_{tst}$ is constant.
>
> 10. **Infinite at c=1**
>
> We first note that our work makes no claims as to what happens exactly at $c=1$.
>
> Second, from the reviewer's comment, we think that the reviewer is using $W = UU^T$ (i.e., the projection on the rank one space). However, in our set up $W = \theta_{trn}X_{trn}Y_{trn}^+ \neq UU^T$. In particular, the exact formula is given in Proposition 2 on page 11. For this, $W$, we see that we have unbounded variance. Empirical verification of Theorem 1 is seen in both Figure 5 (page 8) and Figure 8 (page 33)
>
> Further, in Figure 2b, 2c (page 4), we see a significant increase in the error (empirically as $c \to 1$). Note that the $y$ axis is on a log scale.
>
> 11. **Higher Rank difficulty**
>
> In our setup, $W = \theta_{trn}X_{trn}Y_{trn}^+$ is a random variable that depends on the random variable $Y_{trn} = \theta_{trn}X_{trn} + A_{trn}$ and $A_{trn}$ is a random variable. In such a situation, it is easy to understand the distribution of $A_{trn}$ and hence the distribution of $Y_{trn}$. **However**, we have the pseudo-inverse of $Y_{trn}$ and this is no longer easy to understand. What we would like is to decouple $Y_{trn}^+$ into terms that depend **only** on $A_{trn}^+$ and not on $X_{trn}$ and terms that depend on $X_{trn}$ and not $A_{trn}^+$.
>
> However, the Woodbury formula gives us a term whose inverse involves both $A_{trn}$ and $X_{trn}$, and so it did not help us achieve our goal.

---

### Review · Reviewer_pzdw · 2023-02-25

**Summary Of Contributions:**

Double descent is an interesting phenomenon in overparameterized neural networks and deserves extensive investigations. In this paper, the authors carry out high-quality empirical studies with general input matrices and present some generalization analysis in the case of a rank-1 input matrix and Gaussian noise. This noise setting is interesting. But the network does not involve an activation function.

**Audience:**

Yes

**Broader Impact Concerns:**

It would be more interesting if the authors could extend their theoretical results to more general input matrices and networks with activation functions.

**Claims And Evidence:**

Yes

**Requested Changes:**

The authors should comment on difficulty in extending their theoretical results to more general input matrices and networks with activation functions.

**Strengths And Weaknesses:**

The empirical studies with general input matrices carried out in the paper for double descent and the noise setting are interesting contributions of the paper.
There are two weaknesses. One is the lack of an activation function in the network, and the other is the rank-1 restriction on the input matrix.

---

> ### Author Response · Authors · 2023-02-25
> **Challenges**
>
> The main difficulty in generalizing the work comes from being able to understand the term $W = \theta_{trn} X_{trn} Y_{trn}^+ = \theta_{trn} X_{trn} (\theta_{trn}X_{trn} + A_{trn})^+$. Specifically the $(\theta_{trn}X_{trn} + A_{trn})^+$ term. Individually each term in the pseudo-inverse is easy to understand. However, we have them coupled together. One could look at asymptotic/Taylor expansions of this term, but we have not found one that is accurate when empirically used to approximate the generalization error.
>
> Hence we would like to deal with the term exactly. In the rank one case, there is a formula that is a generalization of the Sherman Morisson formula ($(A+uv^T)^{-1} = A^{-1} - \frac{A^{-1} uv^T A^{-1}}{1-v^TA^{-1}u}$. The important thing is that the inverses are only on the $A$ and not on a coupled term. Hence we can work individually.
>
> As mentioned in the rank one case, such a formula exists for the pseudo-inverse. However, this critical decoupling is no longer easily achieved if we go beyond the rank one case. For example, the generalization of the Sherman-Morrison formula is the Woodbury formula, which states
>
> $$(A+UCV^T)^{-1} = A^{-1} - A^{-1} U(C^{-1} + V^TA^{-1}U)^{-1}V^TA^{-1}$$
>
> Here we have the $(C^{-1} + V^TA^{-1}U)^{-1}$ which is difficult to deal with. This is a scalar and much easier to handle in the rank one case.
>
> Adding non-linearities introduces some issues. First, the problem is no longer convex, and we do not have good descriptions of the global optima. Hence we focused on the first simple case for a linear model. However, we feel these issues might be easier to overcome than the rank one or low-rank assumption.

---

### Public Comment · ~Ding-Xuan_Zhou2 · 2023-02-21
**The linearity of the algorithm and the rank-1 requirement of the input make the paper hard to provide new insights for double descent of neural networks**

The topic of double descent for learning with neural networks is important. The main result of the paper is some asymptotically analysis of the generalization error of a linear algorithm with a rank-1 input in addition to Gaussian type noise. The noise distribution is an interesting point of the paper. But the linearity of the algorithm and the rank-1 requirement of the input make the paper hard to provide new insights for double descent of neural networks. The recent work of Cao, Chen, Belkin, and Gu is also for a special type of noise following a Gaussian distribution on the subspace perpendicular to a signal vector and the neural networks are fully connected ones with weights shared for two classes without convolutions. But their work deals with a nonlinear neural network and can achieve arbitrarily small training error. The authors present some empirical work involving input matrices of rank r>1. But some rigorous analysis needs to be done.

---

> ### Author Response · Authors · 2023-02-21
> **Response**
>
> Thank you for your interest in our work.
>
> Thank you for the pointer to the Cao, Chen, Belkin, and Gu paper. From my understanding, their paper also looks at a signal in dimension 0 subspace (i.e., is $\pm \mu$). *This is even more restrictive than our assumption that the data lies on a line.* Further, their data looks like $x_i^T = [\pm \mu\ \  \xi_i]$ where $\xi_i$ *is perpendicular* to $\mu$ (here $\mu$ is fixed). Hence their noise is represented by extra components in the vector, whereas our noise is isotropic and does not have any relation to the line. Also, the noise is added, not concatenated to the vector. These are major differences.
>
> The paper also looks at the problem of *binary classification*. Which is very different from our denoising setup.
>
> Second, the goal of the paper is as follows:
>
> 1) Present the empirical study showing double descent in the denoising problem for specific networks.
> 2) Presenting the empirical study showing the dependence of the optimal training SNR on the number of training data points.
> 3) In a simple case understanding the phenomena.
>
> **We believe that our paper presents interesting analysis in a new setting. However, to better understand real networks, we agree that more work needs to be done to relax the assumptions.**

---

### Author Response · Authors · 2023-02-25
**Updated Manuscript**

We thank the reviewers for their comments. We are pleased that all reviewers find the work interesting.

The new version has been updated as follows:

1. We have significantly polished the writing.
2. We updated some of the text in Section 2 based on reviewer feedback.
3. Updated the assumptions sections 4.1,4.2,4.3 based on reviewer feedback.
3. Updated Theorem 1.

Please see responses to reviews for more detailed comments. We hope that these responses have addressed all of the reviewer's concerns.

---

### Decision · Action_Editors · 2023-03-21

**Recommendation:** Accept with minor revision

**Comment:**

The paper is on the generalisation performance of denoising models. It demonstrates empirically that under certain conditions, the generalisation error follows a double descent curve and provides a simple model for the observed phenomenon.

This is a borderline paper and the reviewers have mixed opinions about it, 2 leaning to accept while 1 is leaning to reject, one reason being a perceived limited significance.

However all reviewers indicate that the claims are backed up by evidence and that they think there would be some TMLR readers who would be interested in the topic even though the topic and problem setting are rather restricted and a better write-up could make the readers' life much easier.

In the hope that the studied phenomenon and mathematical analysis will be useful for other researchers, I recommend acceptance.

Some comments/questions to address for the camera-ready version:

- p3: typo: rations

- p3: "Our goal is to understand the impact of training noise impacts generalization error"
  impact of training noise on ... ?

- Section 3 intro: ", we explore the role of the amount of training noise and show that
optimally picking theta_tst can mitigate the previously seen double descent."

Is this a typo, and it should be theta_train? Or otherwise, please explain how we can pick theta_test.



**Audience:**

There will be some individuals who will be interested in the findings.

**Claims And Evidence:**

The paper provides empirical evidence and theory providing support of their findings.

---

> ### Author Response · Authors · 2023-04-01
> **Thank you**
>
> We thank the reviewers for their comments and the editor for the feedback and decision. We have uploaded a camera-ready version.
>
> We have done one more thorough read-through to improve clarity by fixing typos. The typos pointed out by the editor have been fixed. The last point brought up by the editor is a typo. We pick $\theta\_{trn}$ and not $\theta\_{tst}$.